# Transcranial photobiomodulation improves insulin therapy in diabetic microglial reactivity and the brain drainage system

Shaojun Liu[1,5], Dongyu Li [2,5], Tingting Yu [1], Jingtan Zhu[1], Oxana Semyachkina-Glushkovskaya [3,4] & Dan Zhu [1✉]

The dysfunction of microglia in the development of diabetes is associated with various diabetic complications, while traditional insulin therapy is insufficient to rapidly restore the function of microglia. Therefore, the search for new alternative methods of treating diabetes-related dysfunction of microglia is urgently needed. Here, we evaluate the effects of transcranial photobiomodulation (tPBM) on microglial function in diabetic mice and investigate its mechanism. We find tPBM treatment effectively improves insulin therapy on microglial morphology and reactivity. We also show that tPBM stimulates brain drainage system through activation of meningeal lymphatics, which contributes to the removal of inflammatory factor, and increase of microglial purinergic receptor P2RY12. Besides, the energy expenditure and locomotor activity of diabetic mice are also improved by tPBM. Our results demonstrate that tPBM can be an efficient, non-invasive method for the treatment of microglial dysfunction caused by diabetes, and also has the potential to prevent diabetic physiological disorders.

[1] Britton Chance Center for Biomedical Photonics—MoE Key Laboratory for Biomedical Photonics, Wuhan National Laboratory for Optoelectronics—Advanced Biomedical Imaging Facility, Huazhong University of Science and Technology, 430074 Wuhan, Hubei, China. [2] School of Optical Electronic Information—Advanced Biomedical Imaging Facility, Huazhong University of Science and Technology, 430074 Wuhan, Hubei, China. [3] Saratov State University, Astrakhanskaya Str. 83, 410012 Saratov, Russia. [4] Physics Department, Humboldt University, Newtonstrasse 15, 12489 Berlin, Germany. [5]These authors contributed equally: Shaojun Liu, Dongyu Li. ✉email: dawnzh@mail.hust.edu.cn

D iabetes mellitus (DM) is one of the most widespread metabolic diseases with a prevalence of 537 million adults worldwide in 2021[1]. The disorder of blood glucose metabolism in patients with DM can induce various complications, including chronic cardiovascular and cerebrovascular dysfunctions, which makes DM the third major killer of humans after cardio-cerebrovascular disease and tumor[2–4]. Diabetic chronic hyperglycemia leads to abnormally high expression of various cytokines and chemokines as well as the development of oxidative stress triggering brain inflammation[5–9].

There is growing evidence that DM-related chronic inflammation can activate microglia, the resident immune cells of the central nervous system (CNS)[10–12]. In vitro studies have shown that under chronic hyperglycemia, microglia become activated showing enlarged cell bodies, enhanced migration, and phagocytosis as well as increased production of pro-inflammatory factors[10,13,14]. However, the exact contribution of microglia to DM-related abnormalities of brain functions remains poorly understood. Current data suggest that DM induces dysregulation of microglial responses to cerebrovascular injuries that are linked to DM-mediated inflammation[15,16]. Chronic hyperglycemia can damage cerebral vessels when the brain receives too little blood leading neurons to die, i.e., brain atrophy, and can cause problems with brain functions, such as memory and thinking[17–19].

Although insulin treatment can effectively control blood glucose levels, it cannot completely restore microglial and brain functions when DM becomes a chronic disease. The tPBM can be an alternative non-invasive method for the therapy of DM-mediated brain injuries, and tPBM is widely used for the treatment of various brain diseases, including intraventricular hemorrhage, Alzheimer and Parkinson's diseases[20–26]. There are also data suggesting that tPBM improves therapy of diabetic foot ulcers and retinal vasculopathy via modulation of the expression of inflammatory factors[27–31]. Recent studies clearly demonstrate that tPBM stimulates the brain drainage system via activation of the meningeal lymphatic vessels (MLVs) and can also have therapeutic effects on brain immunity. Indeed, tPBM can regulate the permeability of the lymphatic endothelium and activate the movement of immune cells in the lymph as well as effectively manage lymphedema[32,33]. The near-infrared photo-effects (1267 nm) stimulate the clearance of toxins from the brain via modulation of the lymphatic tone and contraction[34–36].

In this work, we aimed to study DM-induced changes in microglia. We also hypothesized that tPBM can improve the microglial function in DM mice treated with insulin. First, we established a type 1 diabetic mouse model to track the changes in microglial morphology and the reactivity to cerebrovascular injury during the development of diabetes. Next, we performed tPBM (1267 nm) in diabetic mice treated with insulin, which has been proven effective in treating brain dysfunctions[32,35], and evaluated the improvement of microglial morphology and function. Finally, we studied the mechanisms of the tPBM by evaluating its effects on the blood–brain barrier (BBB) and meningeal lymphatic system in diabetic mice. We found that DM induced changes in microglial morphology and reactivity to cerebrovascular injury. The treatment of tPBM effectively enhanced the therapy of insulin on microglial function in diabetic mice. We also demonstrated that the mechanism of therapeutic effects of tPBM was stimulation of the brain drainage system, which decreased the level of inflammatory factor IFN-γ, and increased the expression of microglial purinergic receptor P2RY12 in brain tissue. Besides, we also found that energy expenditure and locomotor activity in diabetic mice were effectively improved after tPBM combined with insulin treatment.

## Results

**DM induces changes in the microglial morphology.** In the first step, we performed the qualitative and quantitative analysis of the morphological changes of microglia using confocal imaging in the CTR, DM-1W, DM-2W, and DM-3W groups. The results showed that DM did not affect the density of microglia (Fig. 1e). However, the microglial morphology has altered visibly, suggesting microglia responding to DM (Fig. 1a–d). In fact, microglia in the CTR group were in a homeostatic status, with small ellipsoidal bodies and elongated branches. The development of DM over 3 weeks was accompanied by a gradual increase in the soma of microglia and their transformation from regular ellipsoidal shape to irregular ameboid, as well as a reduction in microglial branches (Fig. 1b–d).

The three-dimensional reconstruction and the quantitative analysis of the microglial morphology (Fig. 1) revealed that the average volume of microglial soma was 1.12-fold higher in the DM-1W group vs. the CTR group ($p = 0.003$); 1.06-fold higher in the DM-2W group vs. the DM-1W group ($p = 0.009$); 1.08-fold higher in the DM-3W group vs. the DM-2W group ($p = 0.003$) (Fig. 1f).

The microglial soma sphericity was 1.05-fold lesser in the DM-1W group vs. the CTR group ($p < 0.001$); 1.03-fold lesser in the DM-2W group vs. the DM-1W group ($p = 0.009$), 1.03-fold lesser in the DM-3W group vs. the DM-2W group ($p = 0.009$) (Fig. 1g).

Moreover, the percentage area occupied by branches of microglia dramatically reduced with the DM progression (Fig. 1h). Actually, average percentage of an area occupied by branches of microglia was 1.07-fold lesser in the DM-1W group vs. the CTR group ($p = 0.016$); 1.19-fold lesser in the DM-2W vs. the DM-1W ($p = 0.001$); 1.21- fold lesser in the DM-3W group vs. the DM-2W group ($p = 0.008$) (Fig. 1h).

Thus, these results clearly demonstrate the DM-mediated changes in microglial morphology, suggesting microglia responding to DM in diabetic mice was aggravated with the DM progression.

**DM impairs the microglial reactivity to cerebrovascular injury.** The changes in the microglia morphology inevitably lead to alterations in their functions. Indeed, microhemorrhages in the brain are associated with microglial branches enveloping the injury zone, which can be significantly impaired in DM mice. In order to investigate DM-mediated changes in microglial reactivity to injury of the cerebral vessels, we monitored the dynamic changes in microglial response to photoablation of the cerebral microvasculature in CTR, DM-1W, DM-2W, and DM-3W mice using in vivo two-photon microscopy.

Figure 2a illustrates the design of the microglial response to photoablation of the cerebral microvasculature. Figure 2b shows time-lapse imaging of microglial response to cerebrovascular injury induced by photoablation of the cerebral vessels in different groups. The place of laser application is shown by white arrows in Fig. 2b. The photo-injury was accompanied by an increase in the blood endothelial leakage of the RB, which filled the cerebral vessels. There were visible differences in the microglia reactivity to photo-damages of the cerebral vessels between the healthy and the DM groups. Indeed, in the healthy group, a large number of microglial branches were extended rapidly toward the site of photo-injury, and completely encircled the damaged cerebral vessels 30 min after laser application. However, the microglial reactivity to the cerebral injury was reduced in the DM groups, especially in DM-3W mice. Indeed, in the DM-3W group, the microglial branches were unable to reach the center of the injured vessels 1 h after photoablation, and only a small number

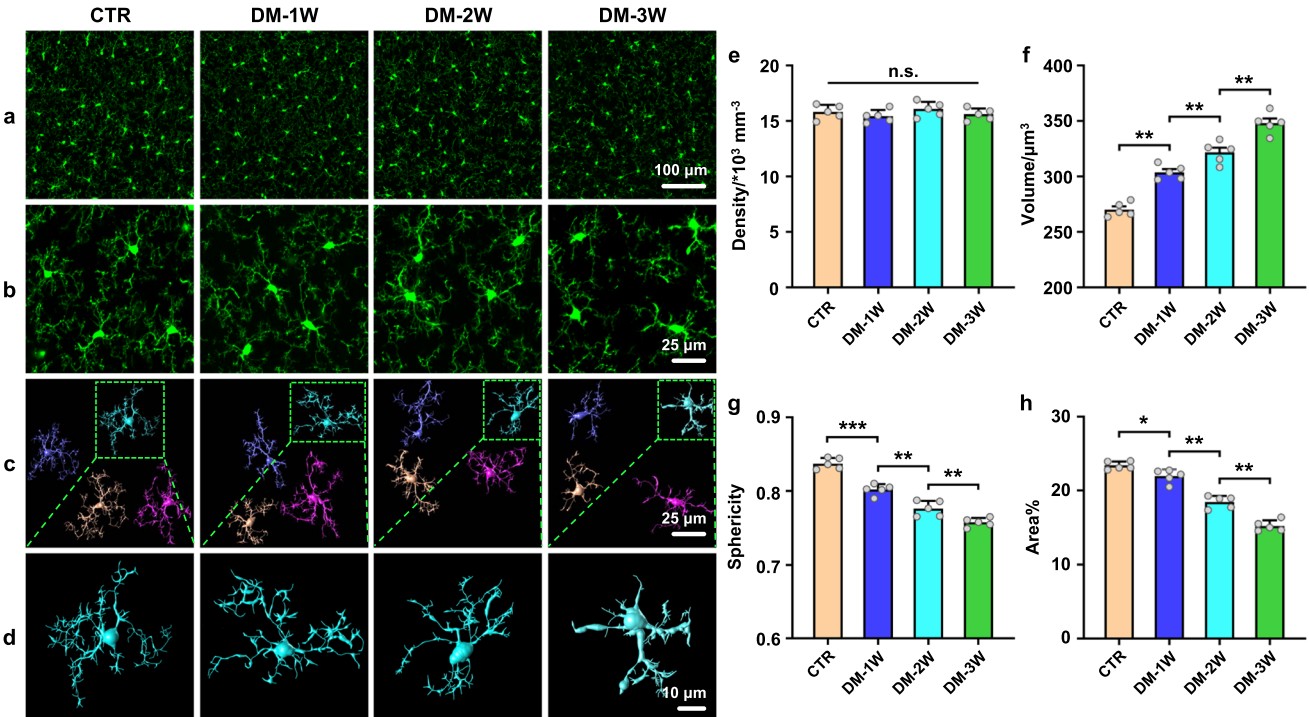

**Fig. 1 Changes of microglial density and morphology induced by diabetes. a** Representative images of the density of cortical microglia in the tested groups. Scale bar: 100 μm. **b** Representative images of morphological changes of microglia in the tested groups. Scale bar: 25 μm. **c** Three-dimensional reconstruction of microglia in the tested groups. Scale bar: 25 μm. **d** Enlarged views of microglia in the green dashed box in (**c**). Scale bar: 10 μm. **e–h** Quantitative analysis of cortical microglial density (**e**), soma volume (**f**), soma sphericity (**g**), and average percentage area of microglia (**h**) in the tested groups. Data were presented as mean ± standard deviation ($n = 5$ mice in each group), and statistical significance in **e–h** was assessed using repeated measurements one-way ANOVA. The statistical tests involved two-sided analyses (n.s., *, **, and *** indicate not significant, $p < 0.05$, $p < 0.01$ and $p < 0.001$, respectively).

of the microglial branches aggregated around the injured site (Fig. 2b, Supplementary Movie 1).

We also performed a quantitative analysis of the changes in the microglial reactivity to cerebrovascular injuries. Two parameters of microglial architecture were assessed, such as the extension velocity of microglial branches, and the accumulation of microglial branches at different distances around the photo-damaged area. The microglial branches closest to the ablation at every 10° angles were detected, and the positions of these branches form the vertices of a polygon that represents a front of microglial branches that will respond to the damage (Fig. 2c). The changes of the polygon area represented the extension velocity of the microglial branches to the photo-damaged area. Besides, several concentric circles spaced 5 μm apart were centered on the photo-damaged area in each image, and the percentage area occupied by microglial branches within each circle reflected the accumulation of microglial branches at different distances from the injury center (Fig. 2d).

Figure 2e clearly demonstrates that in the healthy group, the microglial branches rapidly occupied the damaged area filling the region of interest (ROI) and leading to a decrease in free space of analyzed circles of ROI. A similar microglial reactivity was observed in the DM-1W group. However, there was a gradual reduction of the microglial reactivity to photoablation in the DM-2W and DM-3W groups, respectively. Besides, we also performed quantitative analysis of the time taken for the area of the polygon to be reduced to a minimum (The polygon was considered to have reached its minimum area when it was reduced to less than 10% of its original area) (Fig. 2f). Indeed, the time taken for the area of the polygon to be reduced to a minimum were significantly increased in the DM-2W and DM-3W groups (DM-2W vs. DM-1W, $p = 0.027$; DM-3W vs. DM-2W,

$p = 0.040$), indicating the gradual decrease in the velocity of microglial branches extension.

Supplementary Fig. 1 represents the time-dependent changes of microglial branch accumulation at different distances from the injury in the tested groups. In order to quantitatively analyze the accumulation of microglial branches, we calculated the maximum percentage of microglial branches (Fig. 2g) and the percentage of microglial branches 60 min after the injury (Fig. 2h) at different distances. Figure 2g clearly shows that the maximum percentage of microglial branches in all the distances was higher in the CTR group than in the DM groups, and it also decreased gradually as DM progressed. This suggested that the microglial branches capable of responding to vascular injury were reduced during DM. Besides, there was also a significant reduction of the microglial reactivity manifested by the inability of the microglial branches to reach the center of damage in the DM groups (Fig. 2h). In the healthy group, the microglial branches positioned maximally to the damaged area and occupied 80–90% of the inner region (a distance of 10–20 μm from the center of injury) of ROI 60 min after injury. Conversely, there was only a small number of microglial branches (20%-30%) presented in the periphery (a distance of 30–40 μm from the center of injury) of ROI. Indeed, the percentage of the microglial branches, which were accumulated in the inner region of ROI 60 min after injury was 60–70% in the DM-1W group and 40–50% in the DM-3W group. On the contrary, these groups demonstrate a higher percentage of the microglial branches on the periphery of ROI, especially the DM-3W group (40–50%).

These results clearly demonstrate that DM is associated with a gradual reduction of the microglial reactivity to brain injury induced by photoablation of the cerebral vessels.

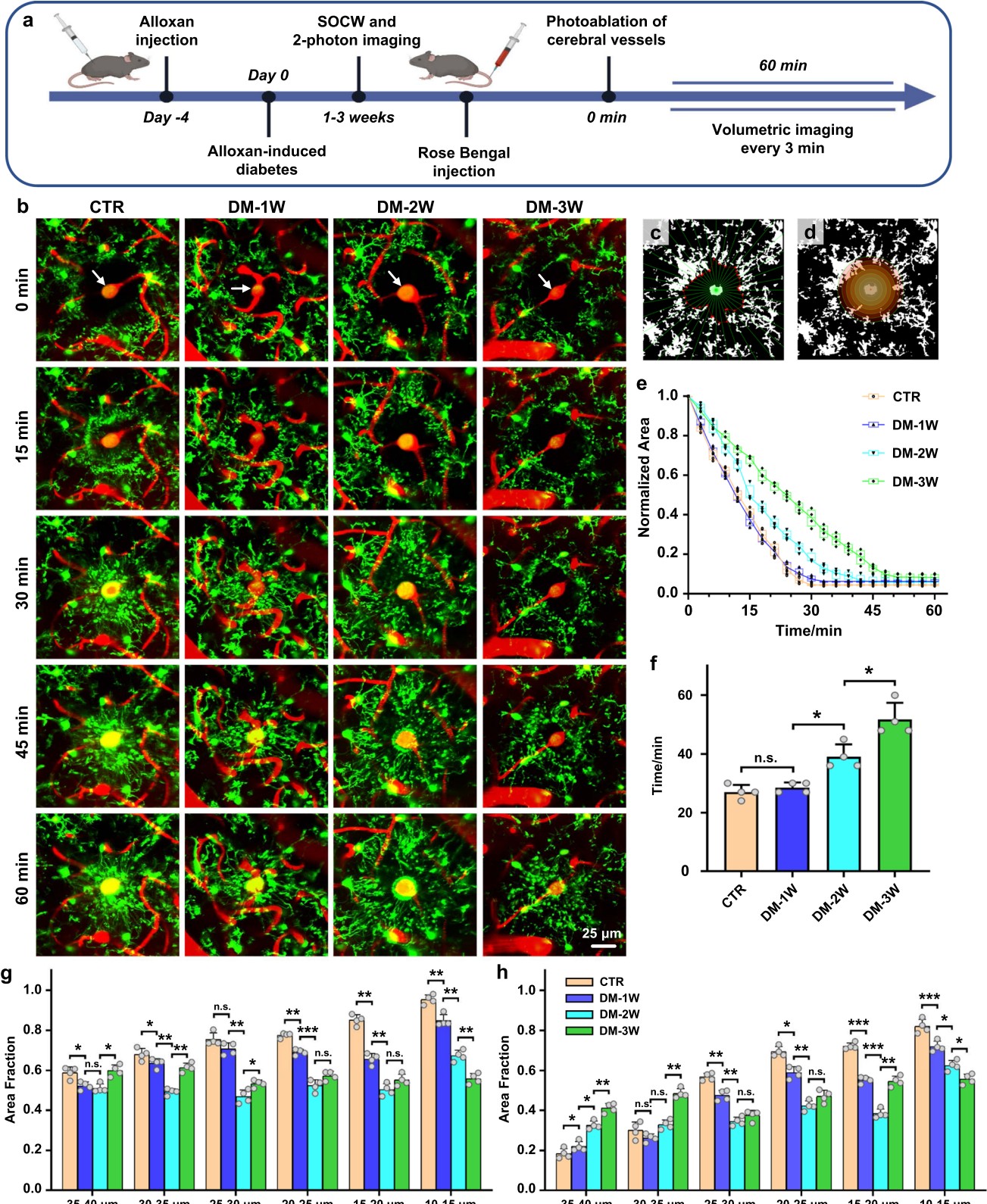

## tPBM improves insulin therapy of microglia in DM mice

To investigate whether tPBM could improve insulin therapy of microglia in DM mice, we studied the effects of tPBM on microglial morphology and reactivity in DM mice received alone (insulin or tPBM treatment) and combined therapy (insulin and tPBM treatment). The design of tPBM treatment is shown in Supplementary Fig. 2. After 7 days of treatment, the changes in microglial density and morphology were analyzed. Since the diabetic mice received 7 days of treatment at DM-2W, the insulin/tPBM effects on microglia were compared with the DM-3W group.

Figure 3 demonstrates the insulin/tPBM effects on the density and microglial morphology followed by three-dimensional reconstruction as well as the quantitative analysis in the tested

**Fig. 2 The changes in the microglial reactivity to photoablation of the cerebral vessels in healthy and diabetic mice. a** The design of the study of the microglial responses to photoablation of the cerebral vessels (SOCW: skull optical clearing window). **b** Representative two-photon microscopic images of the time-dependent changes in the microglia (green) around the photo-damaged area (white arrow) of the cerebral vessels filled by the Rose Bengal (RB, red) in the tested groups. Scale bar: 25 μm. **c** Schematic illustration of the extension velocity of the microglial branches to the photo-damaged area. The green radial lines divide the image into 36 sectors. The vertices of the red polygon correspond to the microglial branches closest to injury in each sector. **d** Schematic illustration of the percentage of microglial branch accumulation within the photo-damaged area presented as circles with a spaced 5 μm. The center of the laser-induced bleeding (0–10 μm) was omitted due to the presence of an autofluorescence signal. **e** The changes of polygon area bound by microglial branches closest to the injury in the tested groups (the box indicates the upper and lower quantiles, the thick line in the box indicates the median and the whiskers indicate 2.5th and 97.5th percentiles). **f** Quantitative analysis of the time taken for the area of the polygon to be reduced to a minimum. **g** Quantification of the maximum percentage of microglial branch accumulation at different distances from the injury in the tested groups. **h** Quantification of the percentage of microglial branches accumulation at different distances 60 min after the injury in the tested groups. Data were presented as mean ± standard deviation (n = 4 mice in each group), and statistical significance (**f–h**) was assessed using repeated measurements one-way ANOVA. The statistical tests involved two-sided analyses (n.s., *, **, and *** indicate not significant, $p < 0.05$, $p < 0.01$ and $p < 0.001$, respectively).

groups. The results showed that both insulin/tPBM alone and insulin + tPBM treatment did not affect the density of microglia (Fig. 3a, e). However, after insulin/tPBM therapy, the volume of microglial soma was 1.15-fold lesser in the DM + I group (DM + I vs. DM-3W, $p < 0.001$) and 1.22-fold lesser in the DM + tPBM group (DM + tPBM vs. DM-3W, $p < 0.001$) (Fig. 3c, d, f). The sphericity of microglial soma and the percentage area occupied by microglial branches were 1.03-fold ($p = 0.018$) and 1.06-fold (0.187) higher in the DM + I group vs. the DM-3W group, 1.03-fold ($p = 0.012$) and 1.26-fold ($p < 0.001$) higher in the DM + tPBM vs. the DM-3W group, respectively (Fig. 3b–d, g, h). The insulin combined with tPBM treatment led to further therapeutic improvement of the microglial morphology. Indeed, the sphericity of microglial soma and the percentage area occupied by microglial branches were 1.05-fold ($p < 0.001$) and 1.36-fold ($p < 0.001$) higher in the DM + I&tPBM group vs. the DM + I group, respectively (Fig. 3b–d, g, h). The volume of microglial soma was also 1.06-fold lesser ($p = 0.022$) in the DM + I&tPBM than in the DM + I groups (Fig. 3c, d, f). Thus, combined tPBM treatment effectively improves insulin therapy, promoting the better recovery of microglial morphology.

To systematically analyze the effects of insulin/tPBM treatment on microglia in diabetic mice, we also evaluated the proliferative properties of microglia and the secretion of cytokines (Supplementary Fig. 3). Here, we used anti-DAPI antibody to stain the nuclear, and the anti-Mki67 antibody was used to stain the proliferative cells in the brain and analyzed the percentages of the proliferative microglia (KI67+ microglia) in different groups. As shown in Supplementary Fig. 3a, there were no KI67+ microglia in all the tested groups, which meant the proliferation of microglia was not changed. In response to changes in the microenvironment of the brain, microglia secrete a variety of cytokines. Herein, we also assessed the changes in the level of several cytokines' mRNA expression. The results in Supplementary Fig. 3b–e showed that mRNA expression of pro-inflammation cytokines (CD86 and TNF-α) was notably increased, while the mRNA expression of anti-inflammation cytokines (CD206 and TGF-β) was decreased in the DM-3W group, both insulin or tPBM alone treatment could reduce the mRNA expression of pro-inflammation cytokines and increase the mRNA expression of anti-inflammation cytokines. These effects are further enhanced by insulin combined with tPBM treatment. These results suggested that more microglia converted towards an anti-inflammatory phenotype after insulin combined with tPBM treatment, to protect against diabetes-induced tissue damage.

In the next step, we analyzed the effects of insulin/tPBM alone and insulin+tPBM on the microglial reactivity. Our results clearly showed that the insulin/tPBM alone treatment improved

microglial response, and DM + I&tPBM group demonstrated a better microglial response to the cerebrovascular injury than the DM + I group (Fig. 4). Both the velocity of microglial branches extension and the accumulation of microglial branches around the center of the injured area was increased after insulin/tPBM and insulin+tPBM treatment. There were more microglial branches that were aggregated around the injured vessel 60 min after photoablation of the cerebral vessels in the DM + I&tPBM group than in the DM + I group (Fig. 4a, Supplementary Movie 1).

The quantitative analysis of the velocity of microglial branches extension and the accumulation of microglial branches around the damaged vessels are shown in Fig. 4b–e and Supplementary Fig. 4. The comparison of the therapeutic effects of insulin/tPBM alone and insulin+tPBM with the DM-3W without therapy revealed a faster response of microglial branches, as the time taken for the area of the polygon bound by microglial branches closest to the injury decresded to a minimun was significantly decreased in all the treatment groups than in the DM-3W group (Fig. 4b, c). Although the velocity of microglial branches extension was not significantly increased in the DM + I&tPBM group than in the DM + I group ($p = 0.952$), the microglial branches capable of responding to vascular injury and the accumulation of microglial branches towards the injured area was significantly increased, as the maximum percentage of microglial branches in almost all the distance was higher in the DM + I&tPBM group than in the DM + I group (Fig. 4d). In addition, there are more microglial branches accumulated around the damaged vessels 60 min after vascular injury in the DM + I&tPBM group than in the DM + I group, especially in the inner (10–20 μm) and surrounding (20–30 μm) area of ROI (Fig. 4e).

Notice, that there were no changes in the microglial morphology and reactivity in healthy mice after the 7 days course of tPBM (Supplementary Fig. 5, Supplementary Fig. 6). Besides, to demonstrate the safety of tPBM, we measured changes in mouse cortical temperature during one course of tPBM in the CTR group. The results showed that the temperature on the cortex surface was increased by only about 0.4 °C during tPBM (Supplementary Fig. 7), which suggested that tPBM did not have a significant thermal effect.

These results suggest that tPBM improves the therapeutic effects of insulin on the DM-mediated changes in the microglial biology and reactivity to cerebrovascular injuries.

In addition to microglia, we also investigated the effects of insulin/tPBM on the physiology of diabetic mice. The energy expenditure, respiratory exchange ratio (RER), and locomotor activity were analyzed to investigate the changes in energy balance during the development of DM in mice (Supplementary Fig. 8). The results revealed that energy expenditure and

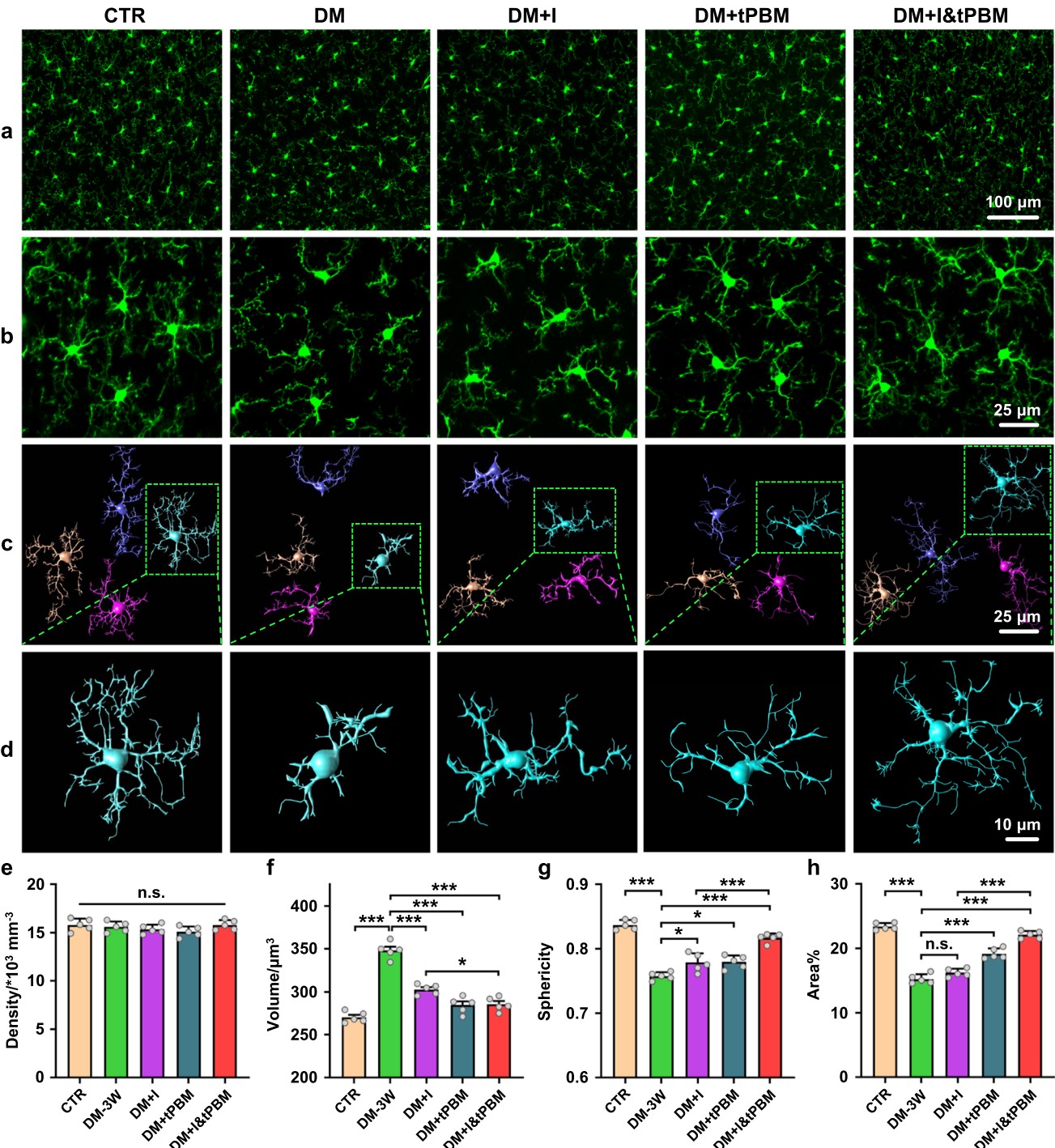

**Fig. 3 Changes of microglial density and morphology induced by diabetes and insulin/tPBM treatment. a** Representative image of the density of cortical microglia in the tested groups. Scale bar: 100 μm. **b** Representative images of morphological changes of microglia in the tested groups. Scale bar: 25 μm. **c** Three-dimensional reconstruction of microglia cells in the tested groups. Scale bar: 25 μm. **d** Enlarged views of microglia in the green dashed box in (**c**). Scale bar: 10 μm. **e–h** Quantitative analysis of cortical microglial density (**e**), soma volume (**f**), soma sphericity (**g**), and average percentage area of microglia (**h**) in the tested groups. Data were presented as mean ± standard deviation ($n = 5$ mice in each group), and statistical significance (**e–h**) was assessed using one-way ANOVA, followed by Tukey's post hoc test. The statistical tests involved two-sided analyses (n.s., *, **, and *** indicate not significant, $p < 0.05$, $p < 0.01$ and $p < 0.001$, respectively).

locomotor activity were significantly reduced that was associated with a decrease in RER in DM-3W mice compared with the CTR group. The combined therapy with insulin and tPBM improved physical activity and changed RER in DM mice. Indeed, energy expenditure, locomotor activity, and RER were approximately identical to those in the control. The separate therapy with insulin or tPBM did not have full therapeutic effects in DM mice. So, insulin increased RER in DM-3W mice, while energy expenditure and locomotor activity did not statistically change from the DM-3W group. The therapy with tPBM did not influence RER and slightly changed energy expenditure and locomotor activity, which were not statistically different from the DM-3W group.

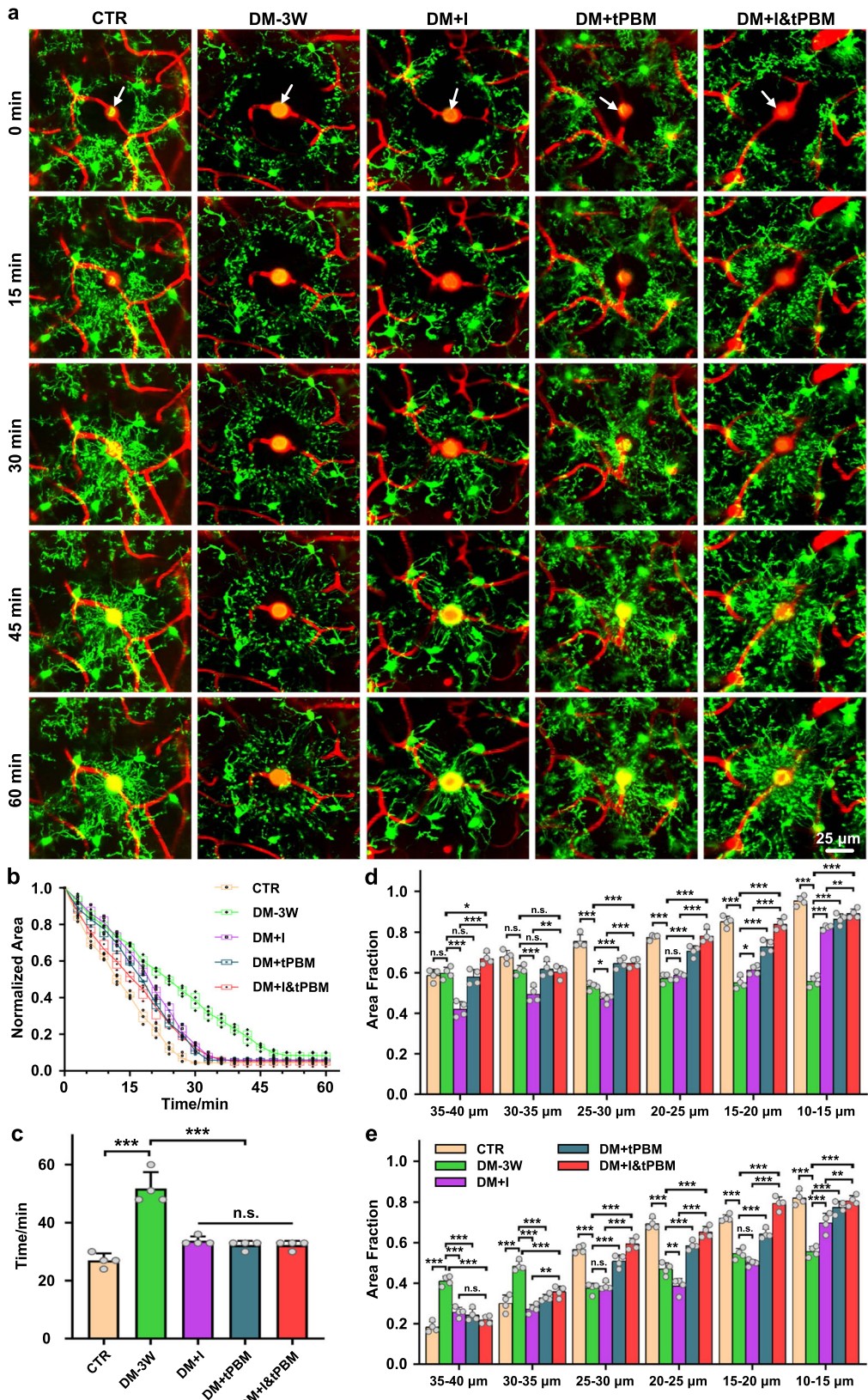

**Mechanisms of therapeutic effects of tPBM**. The DM-mediated changes in the microglial morphology may be accompanied by an increase in the BBB permeability, which exacerbates DM-related cerebrovascular damage and inflammation of the brain tissues[16,37–39]. To study how the BBB permeability changes with the development of DM, we analyzed the leakage of FITC-dextran 70 kDa in the CTR, DM-1W, DM-2W, and DM-3W groups, and investigated the inflammatory status by the measurement of the level of the interferon-gamma (IFN-γ) and the expression of the microglial purinergic receptor P2PY12 in the brain tissues. Supplementary Fig. 9a–c demonstrates that the BBB permeability to FITC-dextran was gradually increased accordingly to DM

**Fig. 4 Quantification analysis of the functional changes of microglia in response to vascular injury in diabetic and insulin/tPBM treatment mice.**
**a** Representative two-photon microscopic images of the time-dependent changes in the microglia (green) around the photo-damaged area (white arrow) of the cerebral vessels filled by the Rose Bengal (RB, red) in the tested groups. Scale bar: 25 μm. **b** The changes of polygon area bound by microglial branches closest to the injury in the tested groups (the box indicates the upper and lower quantiles, the thick line in the box indicates the median and the whiskers indicate 2.5th and 97.5th percentiles). **c** Quantitative analysis of the time taken for the area of the polygon to be reduced to a minimum.
**d** Quantification of the maximum percentage of microglial branch accumulation at different distances from the injury in the tested groups. **e** Quantification of the percentage of microglial branches accumulation at different distances 60 min after the injury in the tested groups. Data were presented as mean ± standard deviation ($n = 4$ mice in each group), and statistical significance was assessed using one-way ANOVA, followed by Dunnett's T3 post hoc test (**c**) and Tukey's post hoc test (**d**, **e**), respectively. The statistical tests involved two-sided analyses (n.s., *, **, and *** indicate not significant, $p < 0.05$, $p < 0.01$ and $p < 0.001$, respectively).

progression and to the increase of blood glucose level. The DM-induced BBB disruption was accompanied by a decrease in the expression of the tight junction (TJ) proteins, ZO-1, and Occludin (Supplementary Fig. 9d, e). These changes were associated with a gradual increase in the IFN-γ level and a decrease in P2RY12 according to the DM development in the DM-1W, DM-2W, and DM-3W groups (Supplementary Fig. 9f, g).

To study the mechanisms of the therapeutic effects of tPBM, we tested the hypothesis that tPBM can promote the recovery of the BBB permeability. Figure 5a, c demonstrates fluorescent microscopy of the BBB permeability to FITC-dextran 70 kDa in ROI and in the whole brain. Our results showed that there was no dye leakage in healthy mice (CTR group) and the high BBB permeability to FITC-dextran in the DM-3W group. There were no changes in the BBB permeability and the expression of ZO1/Occludin after insulin/tPBM and insulin + tPBM treatment, i.e., both therapeutic approaches were not effective for restoration of the BBB integrity despite the recovery of the glucose blood level (Fig. 5a–e). However, after insulin or tPBM alone treatment, the content of inflammatory factor IFN-γ in brain tissue was significantly reduced (1.14-fold lesser, DM + I vs. DM-3W, $p = 0.003$; 1.18-fold lesser, DM + tPBM vs. DM-3W, $p < 0.001$), and the expression of microglial purinergic receptor P2RY12 was 1.08-fold (DM + I vs. DM-3W, $p = 0.236$) and 1.13-fold increased (DM + tPBM vs. DM-3W, $p = 0.014$) (Fig. 5f, g). Moreover, after tPBM combined with insulin treatment, the content of IFN-γ was further reduced (1.10-fold lesser) while P2RY12 was dramatically increased (1.12-fold higher) (DM + I&tPBM vs. DM + I, $p = 0.045$ and $p = 0.016$, respectively) (Fig. 5f, g). This might be responsible for the significant enhancement of microglial response to vascular injury after treatment with tPBM. Considering the important role of IFN-γ signaling in hyperglycemia-induced brain inflammation and impaired microglial response to vascular injury, we concluded the mechanism by which tPBM improved diabetic microglial function was by reducing the level of IFN-γ in brain parenchyma.

In the final step, we analyzed the tPBM-mediated modulation of functions of MLV, which plays an important role in the regulation of brain immunity and clearance of toxins from the CNS[40,41]. First, we studied the effects of a single application of tPBM on the diameter and clearance function of MLVs in diabetic mice. Figure 6a–d clearly showed that the mice in the DM-3W group were accompanied by a significant reduction of lymphatic removal of EBD from the brain and accumulation of dye in the dcLNs due to a decrease in the diameter of the MLVs (16.39 μm for DM-3W vs. 19.69 μm for CTR, $p = 0.001$). However, after the single course of tPBM, lymphatic clearance of EBD was increased via recovery of the diameter of the MLVs to the normal values (20.94 μm for DM + tPBM vs. 19.69 μm for CTR, $p = 0.686$).

Furthermore, we also investigated the effects of a 7-day course of insulin/tPBM and insulin+tPBM on the diameter of MLVs in diabetic mice. Supplementary Fig. 10 showed that only insulin

treatment could not increase the diameter of MLVs in diabetic mice (17.46 μm for DM + I vs. 16.39 μm for DM-3W, $p = 0.727$), while both tPBM and tPBM combined with insulin treatment significantly increased the diameter of MVLs in diabetic mice (20.79 μm for DM + tPBM vs. 16.39 μm for DM-3W, p < 0.001 and 21.51 μm for DM + I&tPBM vs. 16.39 μm for DM-3W, $p < 0.001$). These results suggest that only tPBM has the ability to increase the diameter of MLVs contributing to increased lymphatic function, which was also shown by us earlier[20].

The mechanisms of the tPBM effects on the MLVs remain unclear. There is evidence that it might be via tPBM-mediated stimulation of nitric oxide (NO) production in the lymphatic endothelium[42–45]. To verify the role of NO in the tPBM-mediated stimulation of the MLVs, we investigated the effect of blockage of NO-synthases using L-NAME on the tPBM-mediated changes in the diameter of the MLVs in DM-3W mice. First, we inhibited the release of NO in normal mice with L-NAME and found that the diameter of the MLVs in normal mice was significantly reduced (19.69 μm in CTR group vs. 16.73 μm in CTR + L-NAME group, $p = 0.017$ (Fig. 6c, d)). Then, we also treated DM-3W mice with L-NAME, but the diameter of MLVs in diabetic mice was not changed (16.39 μm in DM-3W group vs. 16.27 μm in DM + L-NAME group, $p = 0.999$ (Fig. 6c, d)). However, our results also showed that there were no changes in the diameter of the MLVs after the single course of tPBM in the diabetic mice treated with L-NAME (17.62 μm for DM + L-NAME+tPBM vs. 16.27 μm for DM + L-NAME, $p = 0.613$ (Fig. 6c, d)), i.e., the NO blockade completely suppressed tPBM-mediated modulation of the MLVs.

These results demonstrate that the dilation of the MLVs due to activation of the NO production plays a critical role in tPBM-mediated modulation of functions of the MLVs.

## Discussion

In our study, we clearly demonstrate the DM-mediated changes in microglial morphology, suggesting microglia responding to DM in diabetic mice that were aggravated with the DM progression. DM is also associated with a gradual reduction of the microglial reactivity to brain injury induced by photoablation of the cerebral vessels. Our results are consistent with data suggesting that DM induces microglial activation and its dysfunction[11,13].

We confirm our hypothesis that tPBM improves the therapeutic effects of insulin on the DM-mediated changes in the microglial morphology and reactivity to cerebrovascular injuries. Indeed, tPBM combined with insulin promotes the better recovery of microglial morphology as well as accumulation of the microglial branches around the damaged cerebral vessels in the DM + I&tPBM group than in the DM + I group.

To study the mechanisms of the therapeutic effects of tPBM, we found that the expression of microglial purinergic receptor P2RY12 was gradually decreased in the different stages of DM. Lou and Haynes et al. demonstrate that the purinergic receptor

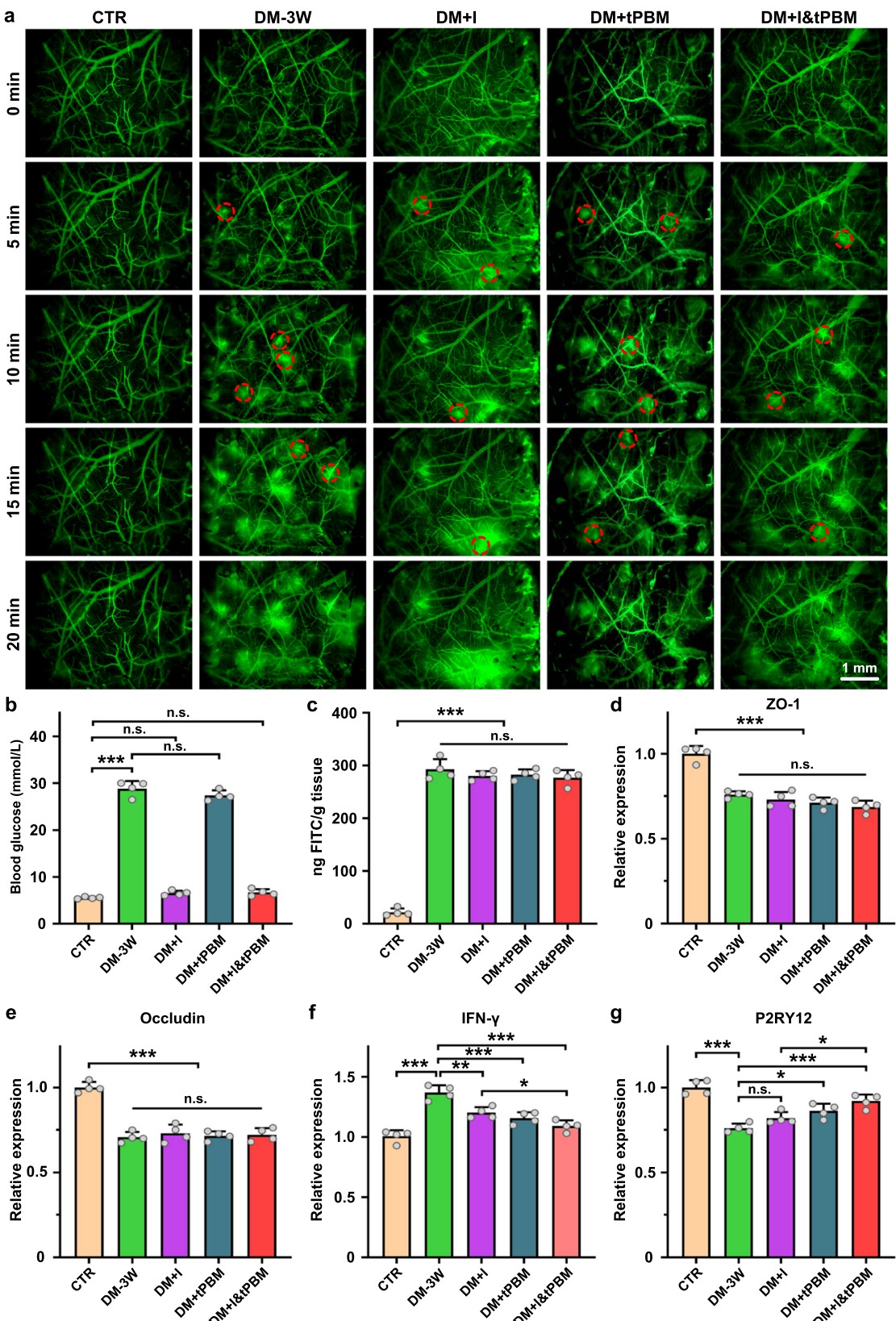

P2RY12 is essential for the ability of microglia to respond to extracellular nucleotides released by brain tissue damage, and the lack of P2RY12 significantly delays the ability of microglia to respond to local brain tissue damage[46,47]. Our data also show a significant decrease of P2RY12 expression in the brain tissues in DM mice that was associated with the impairment of microglial reactivity to cerebrovascular injury. However, both the expression of P2RY12 in the brain tissues and microglia reactivity to cerebrovascular injury were better improved after the combination of insulin with tPBM than insulin therapy alone.

The IFN-γ plays a crucial role in the mechanisms of DM-induced microglial dysfunction[15,48]. Tayler et al. found that four

**Fig. 5 The therapeutic effects of insulin/tPBM on the DM-mediated BBB disruption. a** Time series fluorescence imaging of cortical vasculature after FITC-Dextran 70 kDa injection (The red circles indicate ROI with the location of dye leakage). Scale bar: 25 μm. **b** The change of blood glucose level in the tested groups. **c** Quantitative analysis of FITC-dextran extravasation in the tested groups. **d, e** Changes of the expression of tight junction proteins ZO-1 (**d**) and Occludin (**e**). **f, g** Changes in the expression of proinflammatory cytokine IFN-γ (**f**) and microglial purinergic receptor P2RY12 (**g**) in the brain tissue of the tested groups. Data were presented as mean ± standard deviation ($n = 4$ mice in each group), and statistical significance (**b–g**) was assessed using one-way ANOVA, followed by Tukey's post hoc test. The statistical tests involved two-sided analyses (n.s., *, **, and *** indicate not significant, $p < 0.05$, $p < 0.01$ and $p < 0.001$, respectively).

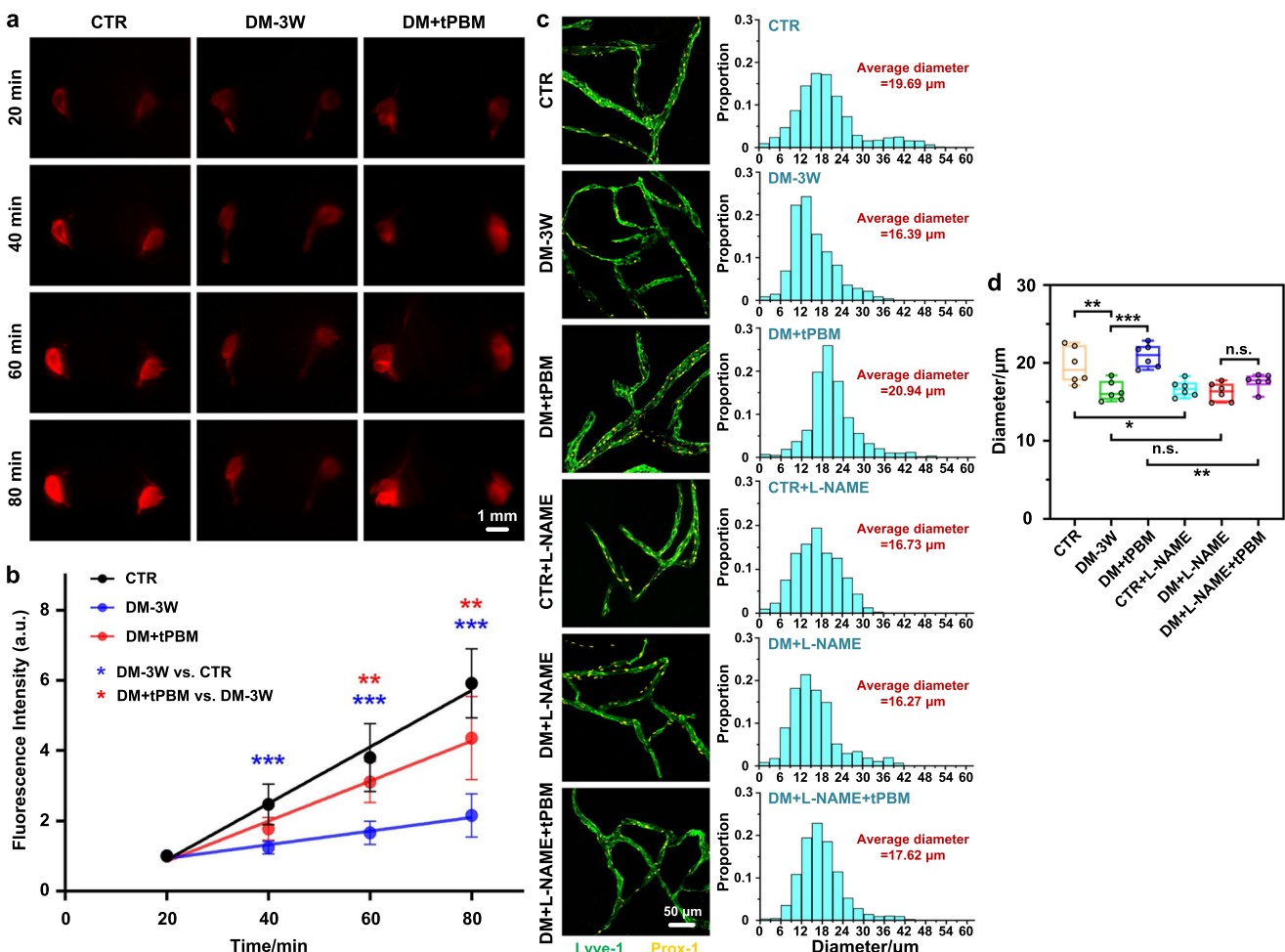

**Fig. 6 Effects of tPBM (single application) on lymphatic removal of EBD and on the diameter of the MLVs. a** Fluorescence images of EBD clearance from the right lateral ventricle into the dcLNs in the tested groups. Scale bar: 1 mm. **b** Quantitative analysis of fluorescence intensity of EBD in the dcLNs of the tested groups. **c** Representative fluorescent images of the basal MLVs (labeled with Lyve-1, green, and Prox-1, yellow) and diameter distribution histogram in the tested groups. Scale bar: 50 μm. **d** Quantitative analysis of the diameter of the MLVs (MLVs: meningeal lymphatic vessels; EBD: Evans blue dye; dcLNs: deep cervical lymph nodes) in the tested groups (the box indicates the upper and lower quantiles, the thick line in the box indicates the median and whiskers indicate 2.5th and 97.5th percentiles). Data were presented as mean ± standard deviation ($n = 6$ mice in each group), and statistical significance was assessed using two-way repeated measurement ANOVA (**b**) and one-way ANOVA followed by Dunnett's T3 post hoc test (**d**), respectively. The statistical tests involved two-sided analyses (n.s., *, **, and *** indicate not significant, $p < 0.05$, $p < 0.01$ and $p < 0.001$, respectively).

weeks after streptozocin-induced DM, the IFN-γ level in the mouse brain tissues is increased significantly and is accompanied by microglia dysfunction[16]. In our study, we also found that the DM-mediated dysfunction of microglia was accompanied by the increase of content of IFN-γ in the brain tissues. Important notice, that only combined therapy of insulin + tPBM was effective for the improvement of this inflammatory factor level and accompanied P2RY12 expression in the brain tissues, while insulin therapy alone was not sufficient. We assumed that tPBM stimulates the lymphatic removal of IFN-γ from the brain tissues leading to a decrease of its level in the CNS. Indeed, we also found that there was a greater increase in the accumulation of IFN-γ in the

dCLNs after combination insulin with tPBM than insulin therapy alone (Supplementary Fig. 11). In the experiments with EBD, we demonstrated that tPBM increases elimination of dye from the brain and its accumulation in the dcLNs that was associated with tPBM-mediated dilation of the MLVs due to activation of the NO production in the lymphatic endothelium. In our previous study, we showed that tPBM effectively stimulates the removal of beta-amyloid from the mouse brain which was accompanied by the improvement of recognition memory in mice with a model of Alzheimer's disease[35,49]. An important role of the MLVs in the regulation of brain immunity and clearance of toxins from the CNS has been shown in other investigations[40,50,51]. In our recent

review, we discussed that tPBM could be a new niche in immu-notherapy of brain diseases, including brain tumors[52].

Diabetes increases BBB leakage due to the loss of TJ proteins, but hyperglycemia and inflammation are not the only damaging factors responsible for increased BBB leakage in diabetes[37]. A large number of studies have shown that diabetes-related hypertension, increased oxidative stress, hyperlipidemia, and insulin resistance contribute to the development of BBB dysfunction[53–55]. Although the tPBM treatment could effectively decrease inflammation in the brain tissue, it needs further investigation whether tPBM has the ability to reverse these changes, increase the expression of TJ proteins, and restore the permeability of BBB. This might explain why our tPBM treatment reduced the level of inflammatory factors but failed to increase TJ protein ZO-1/Occludin expression.

Current studies of the effects of tPBM on the BBB have obtained quite different results. There are data showing that tPBM could increase the BBB permeability by the elevation of reactive oxygen species production and the down-regulation of TJ proteins[34,56,57], and claimed that tPBM would provide new strategies for improving the drug transport across the BBB and improving treatments of brain diseases. However, other studies demonstrated that tPBM could reduce neuroinflammation and protect the BBB in cerebral ischemia or neurotoxin-induced cerebrovascular leakage by the up-regulation of the TJ proteins, revealing the potential of tPBM as a therapeutic tool for reducing vascular dysfunction in neurological conditions[58,59]. In our study on DM mice, neither the leakage of fluorescent dye nor the expression of TJ proteins changed sig-nificantly after tPBM, indicating that tPBM did not evidently influence BBB permeability. Such varying effectiveness of tPBM on BBB might be due to the differences in laser type, laser wavelength, and irradiation dose. Despite multiple encouraging results reported for tPBM in vitro and in vivo, the mechanisms of its therapeutic effect on the brain, especially on the BBB, remain unanswered by researchers. It is commonly accepted that the tPBM mechanisms are mainly based on promoting mitochondrial activity, thereby affecting the corresponding crucial cell functions[60–63]. Therefore, in order to determine the effects of different doses of tPBM on the BBB as well as on the brain, the mechanisms of tPBM require further investigation.

Our results also revealed significant changes in energy balance in DM-3W compared with the control. Indeed, DM was asso-ciated with a decrease in physical activity and energy expenditure that was accompanied by a reduction of RER indicating that fat was the predominant fuel source for DM mice. The combined therapy with insulin and tPBM significantly improved metabo-lism and physical activity in DM mice and it was established by the restoration of the indicated parameters. The separate therapy with insulin or tPBM did not result in full recovery of metabolism and locomotor activity in DM mice suggesting that insulin and tPBM enhance each other's therapeutic effects leading to a sig-nificant improvement of metabolism and energy balance. Our findings are consistent with other results suggesting that tPBM ameliorates curing diabetic complications, such as diabetic foot[64], diabetic periodontitis[65], and diabetic retinopathy[66]. Recent stu-dies indicate that tPBM improves insulin sensitivity in high-fat diet-induced mice[67,68]. TPBM also improves glucose and lipid metabolism disorders in white adipose tissues[67,68] and 3T3-L1 adipocytes[69]. The tPBM-mediated activation of cytochrome c oxidase (CcO)-mediated protein kinase B can be one of the mechanisms of therapeutic effects of tPBM for DM[70]. TPBM accelerates mitochondrial ATP and ROS generation by increasing mitochondrial respiratory chain CcO activity. TPBM-related increase in ATP generation might elevate lipid consumption and attenuate fat deposition in skeletal muscle, whereas the transient ROS production induced by tPBM can promote activation of the PTEN/AKT/GLUT4 and PTEN/AKT/GSK-3β/GS pathways, increasing glucose absorption and glycogen accu-mulation in skeletal muscle.

How tPBM can improve the outcome of DM remains poorly understood. Emerging evidence indicates the important homeo-static and pathophysiological roles of MLVs in the progression of various cerebral small vessel diseases, including DM[71–74]. Indeed, genetic and pharmacological disruption of MLVs results in less drainage of CSF and ISF to dcLNs[36,75]. Such disruption also results in cognitive impairment and behavioral alterations due to dys-functions of the brain's waste removal system leading to the accumulation of toxins and metabolites in the CNS[75]. The dys-function of the MLVs might provide an important contribution to age-related cognitive decline and neuro-degenerative disease[76,77]. The MLVs also affect microglia responses[75]. In normal conditions, MLVs naturally contain a substantial pool of immune cells, such as T cells, macrophages, dendritic cells, mast cells, neutrophils, monocytes, macrophages, and B cells[78–80]. Under brain pathology, these immune cells infiltrate the brain parenchyma from CNS borders through the direct vascular-like channels and further trigger a series of immune responses of neurons and microglia[81]. For instance, T cells in the meninges modulate behavior and cog-nition by releasing cytokines in mice[82]. Emerging evidence has highlighted the capacity of hematogenous cells in skull bone marrow to enter the meninges via ossified vascular channels and maintain immune homeostasis in the CNS. CNS-adjacent skull bone marrow comprises hematopoietic niches that can sense CNS injury and supply specialized immune cells to fine-tune inflam-matory responses. From this aspect, the brain drainage system, including MLVs, is considered a messenger regulating neuroin-flammation and neuroimmune[71,83]. Interestingly, the stimulation of meningeal lymph angiogenesis via administration of the vascular endothelial growth factor can improve the drainage of macro-molecules to the cervical lymph nodes leading to an increase in resistance to brain diseases[75,84–87]. Thus, tPBM-mediated aug-mentation of the MLVs functions, which we showed in our pre-vious works also[20], can be one of the mechanisms of improvement of therapeutic effects of insulin, including microglia reactivity as a part of tPBM-related regulation of neuroimmune providing restoration of homeostasis in the CNS thereby improving recovery of energy balance in DM + I mice.

In conclusion, we clearly demonstrate that DM induces changes in microglial morphology and reactivity to cere-brovascular injury that is associated with BBB leakage and inflammation of the brain tissues. The treatment of tPBM can effectively enhance the therapy of insulin on microglial mor-phology and function in diabetic mice. The mechanisms of therapeutic effects of tPBM are stimulation of the brain drainage system, which improves brain immunity through a decrease in the level of inflammatory factor IFN-γ, and an increase in the expression of microglial purinergic receptor P2RY12 in brain tissues that we observed in DM mice treated with tPBM+insulin. Given the important role of microglia in brain homeostasis and development, restoring microglial function may have the poten-tial to ameliorate diabetic brain dysfunction as microglia are functionally abnormal during diabetes mellitus. But this needs to be further investigated in future work. Besides, we also found tPBM treatment could effectively improve the energy expenditure and locomotor activity in diabetic mice. These results demon-strate the potential of tPBM to treat diabetic microglial dys-function and prevent physiological disorders in diabetes.

## Methods

**Animals**. Adult Balb/c and *CX3CR1*^EGFP/+ mice (2–3 months old) were used in this study. Balb/c mice were supplied by the

Wuhan University Center for Animal Experiments (Wuhan, China). Transgenic mice expressing enhanced green fluorescent protein in microglia ($CX3CR1^{EGFP/+}$) used for imaging microglia were purchased from the Jackson Laboratory (Bar Harbor, ME, USA). All animals were housed in the Wuhan National Laboratory for Optoelectronics at a specific pathogen-free facility, with standard food and water provided *ad libitum*, with a normal cycle (12 h light/dark). All animal procedures were approved by the Experimental Animal Management Ordinance of Hubei Province, P. R. China, and carried out in accordance with the guidelines from the Huazhong University of Science and Technology and were approved by the Institutional Animal Ethics Committee of Huazhong University of Science and Technology. We have complied with all relevant ethical regulations for animal use.

**Model of induction of DM in mice**. Type 1 DM was induced in 2–3 months Balb/c and $CX3CR1^{EGFP/+}$ mice by subcutaneous injection of 150 mg/kg alloxan monohydrate (30 mg/ml, Sigma-Aldrich, USA) for four days after 4 hours fasting every day. The monitoring of the blood glucose was performed to validate the type 1 DM mice model, using an Accu-Chek Active blood glucose meter (Roche Co., Germany) by a drop of blood withdrawn from the tail vein. Afterward, type 1 DM mice were divided into four groups depending on DM progression, namely DM during 1–3 weeks (DM-1W, DM-2W, and DM-3W, respectively) mice used for experiments. The control group (CTR) included intact mice.

**Insulin administration**. To control blood glucose levels in a subgroup of DM mice, insulin (0.003 U/g, i.p., Wanbang Pharmaceutical Co., China) was administered in DM-2W mice three times a day (10:00 am; 16:00 pm; 22:00 pm) for one week ("DM + I" group). Afterward, the blood glucose was monitored.

**Transcranial photobiomodulation**. Supplementary Fig. 2 illustrates schematically the use of tPBM in mice. The mouse was fixed in the stereotaxic frame under inhalation anesthesia (1% isoflurane at 1 L/min $N_2/O_2$—70/30 ratio) and hair was removed from the heads. The irradiation source was a wavelength-locked fiber Bragg grating with a laser diode emitting at 1267 nm (LD-1267-FBG-350, Innolume, Dortmund, Germany). A single-mode distal fiber was connected to the laser diode and terminated by collimation optics to provide a beam diameter of 5 mm. The sagittal sinus area on the head of the mouse was irradiated with a laser dose of 32 J/cm$^2$ for 7 days in the following sequence: 17 min—irradiation, 5 min—pause, 61 min in total[20,35]. During the experiments, the body temperature of the mouse was maintained at 37 °C with a heating pad.

The course of tPBM for 7 days was performed in the following groups: (1) CTR (CTR + tPBM); (2) DM-2W (DM + tPBM); (2) DM + I (DM + I&tPBM). We used the group of DM-3W as a comparison because the insulin/tPBM treatment was performed during 7 days in the DM-2W and when we analyzed the microglial morphology and function, these mice were already DM-3W.

The summary of all test groups and treatment durations is shown in Supplementary Fig. 12. Actually, the mice until the time of treatment were $n = 16$ in number, in a homogeneous group and then divided into 4 groups of 4: DM-3W, DM + I, DM + tPBM, DM + I&tPBM.

**Measurement of cortical temperature changes during tPBM**. To measure the changes in brain surface temperature during the tPBM, the mouse was fixed in the stereotaxic frame under inhalation anesthesia (1% isoflurane at 1 L/min $N_2/O_2$—70:30).

The skin of the head was cut from the right side and the medial part of the right temporal muscle was detached from the skull, then a small hole was drilled in the temporal bone with a cranial drill and a flexible thermocouple probe was inserted into the epidural space between the parietal bone and the brain. Brain surface temperature was measured before and during the laser stimulation in 10 s increments using a handheld thermometer system (CAIPUSEN, YET-620L, China).

**Assessment of the BBB leakage**. In vivo, monitoring of BBB leakage was administrated by a stereo fluorescence microscope (Axio Zoom. V16, Zeiss, Jena, Germany). For visualization of cortical vessels with high resolution, a skull optical clearing window (SOCW) was established before imaging[88–91]. The process was briefly described as follows, the scalp of anesthetized Balb/c mouse was shaved and a midline incision was made on the scalp along the direction of the sagittal suture. Then, a holder was glued onto the skull, and the mouse was fixed with a custom-built immobilization plate. The skull optical clearing treatment was performed in two steps. The reagent S1 was applied to the exposed skull for 10 min, then, the reagent S2 was treated for 5 min after removal of S1. After the establishment of SOCW, 125 µL of FITC-Dextran (Aladdin, China) solution (1%, w/v) with a molecular weight of 70 kDa was used to label blood vessels by inner canthus injection. After that, fluorescence imaging was performed to monitor FITC-Dextran leakage across the BBB for 20 min.

In order to quantify the leakage of FITC-Dextran, we performed transcardially phosphate-buffered saline (PBS) perfusion 20 minutes after FITC-Dextran injection to wash out the fluorescent dye from the vessels, and the dye leaking from the BBB would remain in the brain parenchyma. Then, the isolated brain tissue was cut and weighed, followed by homogenization in 1 mL of 50% trichloroacetic acid (Aladdin, China), using an electronic homogenizer. The homogenate was further centrifuged for 20 min at 10,000×g. After that, the supernatant was aspirated with a capillary glass tube, and the intensity of the fluorescence signal was measured by confocal microscopy[92,93] (Supplementary Fig. 13a).

For measuring the concentration of the fluorescent signal in the supernatant, a standard curve of FITC–Dextran concentration and fluorescence intensity needs to be established. For this purpose, we configured several standard concentrations of FITC-Dextran solutions, including 10, 50, 100, 200, 500, and 1000 ng/mL. Then, we measured the fluorescence intensity at each concentration using confocal microscopy, and fitted a standard curve between FITC-Dextran concentration and fluorescence intensity (Supplementary Fig. 13b), the formula was as follows:

$$I = 0.077 * e^{2.5675*(\lg C)} \tag{1}$$

Where $I$ is fluorescence intensity, $C$ is the concentration of FITC–Dextran (ng/mL). Then, we can calculate the concentration of FITC-Dextran in the supernatant, and quantify the leakage of FITC–Dextran in each brain tissue.

**Ex vivo brain slice preparation and imaging**. To analyze the morphological changes of microglia, $CX3CR1^{EFFP/+}$ mice in each experimental group were perfused and fixed by PBS and 4% paraformaldehyde (PFA, Sigma-Aldrich, USA), then the brain was taken out and placed into 4% PFA overnight for post-fixation. The brain was then sliced (100 µm) with Vibratomes (VT1000, Leica, German) and imaged by confocal microscopy (LSM710, Zeiss, Germany). Image stacks of microglia at the brain cortex were collected at 2 µm z step size by 20× and 63× objective

lens in order to count the density and morphology of microglia, respectively.

**Two-photon imaging and induction of vascular injury**. For visualization of cortical microglia with high resolution, we established SOCW before two-photon imaging, which was as previously described[88]. After establishing SOCW, we performed retro-orbital injection of Rose Bengal (RB, 1% solution, 60 mg/kg, Sigma-Aldrich, USA) to label cerebral vessels[94]. The vascular injury was induced by focusing the high-power (~60–80 mW) two-photon laser beam (900 nm) on the microvessel (5–10 μm in diameter) for 8–10 s. The vascular damage was evaluated as the appearance of extravasation of fluorescence. Subsequent fluorescent images were acquired for the detection of microglial responses to injury. The image stacks were collected every 3 min for 60 min using laser scanning microscopy (Nikon Ni-E, Japan) coupled with a Mai Tai HP Deepsee Laser. Three-dimensional image stacks were obtained using a Nikon 25× 1.1NA water immersion lens at a zoom of 3.2. The emission laser was 920 nm, and image stacks were collected at 2 μm z step size with an area of $133 \times 133$ μm ($1024 \times 1024$ pixels, 0.13 μm/pixel).

**Analysis of microglial morphology and dynamics**. Confocal imaging stacks of microglia in brain slices were processed with Imaris v9.0 software (Bitplane AG, Switzerland). After rendering microglial soma, the density of microglia, the volume and sphericity of microglia soma were counted to analyze microglial morphological changes. Then projecting image stacks along the z-axis by ImageJ software, segmenting images, and calculating fluorescence signal coverage area to quantify microglia occupied area and normalized by the whole image filed.

A custom-written code implemented in Matlab (MathworksTM, Natick, MA), was used to measure the radial movement of microglial branches in response to vascular damage. Initially, time-course microglia image stacks were processed using ImageJ software. Three images above and below the center of the injury (7 images total, z depth = 14 μm) were projected. Then, images from all time points were filtered and corrected for any X–Y displacement by the Matlab code. After binarization of 2-dimension time series images, microglial branches closest to the center of the injured blood vessel were detected at every 10° angle. The positions of these branches form the vertices of a polygon that represents a front of microglial branches that will respond to the damage. The size of the area surrounding the injury bound by the polygon was calculated at each time point. To measure the accumulation of microglial branches around the injury, several concentric circles spaced 5 μm apart were centered on the injury in each image in the time series. The aggregation of microglial branches in different circles was quantified by binarized signal area, and normalized by the corresponding circle area. The innermost ring (0–10 μm) was excluded from analysis due to the appearance of autofluorescence from the laser-induced vascular injury.

**Fluorescence microscopy monitoring of Evans blue accumulation in dCLNs**. Anesthetized mice were fixed on a stereotaxic frame, cutting the skin of the head to expose the skull. A small hole was made in the right side of the skull (AP = −0.5 mm, ML = −1.06 mm) with a cranial drill. Afterward, we injected 5 μL Evans blue dye (EBD, 1% v/v, Sigma-Aldrich) into the right later ventricle to a depth of 2.5 mm, at the speed of 0.5 μL/min. About 20 min later, the skin of the mouse's neck was cut and the muscle was separated laterally to expose the deep cervical lymph nodes (dCLNs). Then, the aggregation of EBD in the dcLNs without and with tPBM was imaged by a stereo fluorescence microscope (Axio

Zoom. V16, Zeiss, Jena, Germany). After imaging, ImageJ software was used to measure the fluorescence intensity of EBD.

**L-NAME treatment**. To inhibit the release of NO, treatment with L-NAME (the blocker of NOS) was performed according to previous publications[20,95,96]. Briefly, the mouse was anesthetized with ketamine hydrochloride and the head was fixed in the stereotaxic frame. And 5 μL (100 mg/mL) L-NAME solution (Sigma, Cat. No. N5751) was injected into the cisterna magna over a period of 5 min. Four hours later, the mouse was used to analyze the changes in MLVs diameter.

**Analysis of energy and locomotor activity**. The measures of metabolism such as energy expenditure and respiratory exchange ratio (RER) as well as locomotor activity were simultaneously measured for each mouse after 10 day adaptation period using an infrared sensor-based technology (PhenoMaster TSE Systems, Supplementary Fig. 8a). The PhenoMaster system automatically screens behavioral and metabolic parameters in a home-cage like environment with a high temporal and spatial resolution. Data are continuously collected from single experimental animals under conditions that maximize the likelihood of natural behavior. The calorimetric parameters were measured indirectly based on conventional Sealsafe Plus GM500 cages equipped to individually monitor one mouse per cage. RER, a measure of metabolism substrate choice (carbohydrate or fat), was calculated as the ratio between $VCO_2$ and $VO_2$. Locomotor activity is measured by a light beam-based device, in which two infrared sensor frames lay on top of each other and surround the home cage. Locomotor activity was determined by the ambulatory movements which were counted as the consecutive interruption of different beams. The minimal motions (such as breathing) were omitted. The mice were monitored over a 24-h period without changing the normal habitat of the animal.

**Western blot**. In order to explore the reasons for dysfunction of BBB and microglial reactivity, the related proteins were extracted from fresh ex vivo brain tissue of mice. The total protein of each sample was measured by the BCA protein assay kit (Thermo Scientific, Rockford, USA). The protein from each sample was separated by 4–12% NuPAGE (180-8018H, Tanon). After transfer of proteins to polyvinylidene fluoride membrane (Millipore, Billerica, USA), the membrane was blocked with 5% skim milk for 1.5 h at room temperature and incubated for 2 h at room temperature with the primary antibodies: anti-ZO-1 antibody (1:500, ab96587, Abcam, Cambridge, United Kingdom), anti-Occludin (1:1000, ab216327, Abcam, Cambridge, United Kingdom), anti-IFN-γ (1:500, 16731181, Invitrogen, Molecular Probes, Eugene, Oregon, USA), and anti-P2RY12 (1:2000, ab184411, Abcam, Cambridge, United Kingdom). After that, the membrane was incubated with HRP-conjugated secondary antibody at room temperature for 1 h and treated with an enhanced chemiluminescent reagent kit (Thermo Scientific, Rockford, USA). The bands were scanned and digitalized, and the density of each band was quantified using ImageJ software and normalized to the values of β-actin. Original scanned images of the blots and gels are included in Supplementary Fig. 14.

**Quantitative reverse transcriptase-PCR**. Total RNA was extracted from the brain of mice using the RNAprep Pure Tissue Kit (DP431, Tiangen Biotech Co., Ltd, Beijing, China) and converted to cDNA utilizing the HiFiScript cDNA Synthesis Kit (CW2569, CoWin Biosciences, China). The ChamQTM Universal SYBR® qPCR Master Mix (Q711-02, Vazyme, China) was used to perform quantitative real-time PCR and then analyzed

with the QuantStudioTM Design & Analysis Software (Thermo Fisher, Waltham, MA, USA). For the analysis, each sample was scrutinized three times to reduce experimental errors. The expression value of each mRNA was normalized to GAPDH mRNA. The primer sequences used for PCR were as follows: 5'-CCCCAAAGGGATGAGAAGTTC-3' and 5'-CCTCCACTTGGT GGTTTGCT-3' for TNF-α; 5'-CTCCCGTGGCTTCTAGTGC-3' and 5'-GCCTTAGTTTGGACAGGATCTG-3' for TGF-β; 5'-TC CAGAACTTACGGAAGCACCCACG-3' and 5'-CAGGTTCACT GAAGTTGGCGATCAC-3' for CD86; 5'-CTTCGGGCCTTT GGAATAAT-3' and 5'-TAGAAGAGCCCTTGGGTTGA-3' for CD206; 5'-AACGACCCCTTCATTGAC-3' and 5'-GAAGACA CCAGTAGACTCCAC-3' for GAPDH.

**Immunofluorescence**. To visualize meningeal lymphatic vessels (MLVs), anti-Lyve-1 and anti-Prox-1 antibodies were used to label MLVs. To collect the meninges, mice were sacrificed, the skin was removed from the head, and the muscles were stripped from the bone. After removing the mandibles and the skull rostral to maxillae, the top of the skull was removed with surgical scissors. Whole-mount meninges were fixed while still attached to the skull cap with 4% PFA overnight at 4°C. The meninges were then dissected from the skull. The meninges were incubated in the blocking solution (a mixture of 0.2% Triton-X-100 and 10% normal goat serum in PBS) for 1 h, followed by incubation with Alexa Fluor 488-conjugated anti-Lyve-1 antibody (1:500; FAB2125G, R&D Systems, Minneapolis, Minnesota, USA), rabbit anti-Prox-1 antibody (1:500; ab101851, Abcam, Cambridge, United Kingdom) overnight at room temperature. Then, the meninges were incubated with Alexa Fluor 555 goat anti-rabbit IgG (H + L) (1:500, A21429, Invitrogen, Molecular Probes, Eugene, Oregon, USA). According to the above steps, the slices of dCLNs of mice were also incubated by Alexa Fluor 488-conjugated anti-Lyve-1 antibody (1:500; FAB2125G, R&D Systems, Minneapolis, Minnesota, USA) and anti-IFN-γ (1:500, 16731181, Invitrogen, Molecular Probes, Eugene, Oregon, USA), followed by incubation with Alexa Fluor 647 goat anti-rabbit IgG (H + L) (1:500, A21245, Invitrogen, Molecular Probes, Eugene, Oregon, USA). And 100 μm brain slices were incubated by anti-Mki67 (1:500 dilution; K009725P, Solarbio®, Beijing, China), followed by incubation with Alexa Fluor 555 goat anti-rabbit IgG (H + L) (1:500 dilution; A21429, Invitrogen, Carlsbad, CA, USA). DAPI (1:1000 dilution; D1306, Invitrogen, Carlsbad, CA, USA) was used for nuclei staining. After that, immunostained tissues were imaged using a confocal microscope (LSM 710, Zeiss, Germany) and analyzed with ImageJ software.

**Measurement of the meningeal lymphatic vessel diameter**. A custom-written code implemented in Matlab was used to measure the diameter of MLVs. Briefly, Otsu's method was used to decide the threshold and obtain the binary image. Then, the broken edges of the image were connected, the holes were filled and the small connected domains were removed. Finally, the obtained image was subtracted by itself after morphological corrosion, and the profile curve was then determined. After that, the diameter distribution of lymphatic vessels was calculated. Choose a point on each side of the lymphatic vessels, and when the tangent lines at both points are perpendicular to the line between the two points, the line between the two points can be taken as the diameter of the lymphatic vessels at this location. Following the above rule, we could obtain a series of lymphatic vascular diameters at every position.

**Statistical and reproducibility**. Statistical analysis was performed using the SPSS (IBM) software. Sample sizes were presented in the figure legends, and the $n$ value was defined as the number of independent animals. Data were presented as the mean ± standard deviation. The normality of data distribution was assessed using the Shapiro–Wilk test, a method for small sample sizes data. Then, the heterogeneity of variance was evaluated using the Levene test. The significance of differences was evaluated by unpaired Student's $t$ test (normality distribution, variance homogeneity) for two independent group comparisons. One-way ANOVA with Tukey's multiple-comparison test (variance homogeneity) or Dunnett's T3 multiple-comparison test (variance non-homogeneity) was used for comparisons of more than two groups. Repeated measurements ANOVA was used when analyzing the changes with the development of diabetes, and non-repeated measurements ANOVA was used when analyzing the changes induced by insulin/tPBM and insulin + tPBM treatment. All experiments were repeated at least twice independently with similar results. All $p$ values < 0.05 were considered statistically significant (*$p < 0.05$, **$p < 0.01$, and ***$p < 0.001$).

**Reporting summary**. Further information on research design is available in the Nature Portfolio Reporting Summary linked to this article.

## Data availability
The source data for the graphs in this study are provided in Supplementary Data 1. All other supporting data of this study are available from the corresponding author upon reasonable request.

## Code availability
The codes used to analyze the function of microglia and the diameter of meningeal lymphatic vessels are freely available on GitHub.

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

## Acknowledgements

This work was supported by the National Natural Science Foundation of China (NSFC) (Grant Nos. 61860206009, 62375096, 82372012, 62375095, and 82001877); Key Research and Development Project of Hubei Province (No. 2022BCA023); Hubei Provincial Natural Science Foundation of China (2023AFB861); the Innovation Fund of WNLO; RF Governmental Grant (No. 075-15-2022-1094). The authors also thank the Optical Bioimaging Core Facility of WNLO-HUST for support in data acquisition.

## Author contributions

S.L. and D.L. were involved in the conceptualization, experiment setup, investigation, statistical analysis, writing, and editing. T.Y. and J.Z. were involved in experiment setup and investigation. O.S. was involved in writing and editing. D.Z. was involved in writing, editing, conceptualization, and project management. All authors read and approved the final paper.

## Competing interests

The authors declare no competing interests.

## Ethical approval

All animal procedures were approved by the Experimental Animal Management Ordinance of Hubei Province, China, and carried out in accordance with the guidelines for the humane care of animals.
