## [Peer Review File · Communications Biology]

Transcranial photobiomodulation improves insulin therapy in diabetic microglial reactivity and the brain drainage systemReviewers' comments:

Reviewer #1:

Remarks to the Author:

The current manuscript by Liu et al., is a rigorous and elegant study where the authors analyze the potential beneficial effect of PBM on microglial morphology and brain drainage system using an Alloxan-induced model of diabetes mellitus. The use of various methodologies and multidisciplinary approaches allow robustness to the conclusions.

The manuscript can be endorsed for publication in its current state by responding to the following minor comments

It is not clear why figure 5 does not include the groups DM-2W+I-1W and DM-2W+I&tPBM-1W as in the previous figures.

In addition, representative fluorescent images of the basal MLVs (labeled with Lyve-1, green and Prox-1, yellow) and diameter distribution histogram in the control group with only L-NAME treatment should be included.

It is not clear in the discussion, why PBM decreases inflammation (relative expression of IFN) but the BBB remains permeable and does not reverse the changes in the expression of ZO1/Occludin as in the DM-3W group.

Reviewer #2:

Remarks to the Author:

The study is original and presents interesting results to better understand the influence of tPBM and the cerebral mechanisms of action involved, in the context of the treatment of diabetes. The article is well written, researched and accurate.

It would be interesting to specify why this diabetic mouse model was chosen, by comparing with the existing one in the literature. Such as for the choice of the wavelength of the tPBM. Could you also specify the local increase in temperature, at the level of the tissue, if there is one? Finally, could you specify the time slot for tPBM treatment? Note that the method of anesthesia is also likely to affect the functioning of the BBB.

Regarding the results. The study is complete; nevertheless the presentation of the graphs and their naming is not implicit easily. A summary figure of the groups and durations of treatment, and comparisons made will facilitate the understanding of the article.

The number of 4 animals per group is low, but nevertheless allows us to observe significance, a sign that the effect of the treatments is indeed present. Nevertheless, concerning the statistical analysis of the data, it should be indicated what type of ANOVA is carried out, and whether the measurements are repeated or not. The mice until the time of treatment should be n=12 in number, in a homogeneous group (box plot) and then divided into 3 groups of 4: DM-3W, DM-2W+I-1W, DM-3W+ I-1W+tPBM.

Finally, the discussion is original and well constructed.

Reviewer #3:

Remarks to the Author:

In the current manuscript, Liu et al. aimed to search for new methods to restore the dysfunction of microglia during diabetes and to improve the pathology. They first established the diabetes model in mice through alloxan injection, and described the changes in morphology and motility of microglia that followed the progression of disease. Next, using transcranial photobiomodulation with 1267nm near infrared light, they demonstrated the beneficial effect on reversing microglia

dysfunction, which correlated with the changes of P2Y₁₂R level. Finally, they checked the brain lymphatic system after tPBM and found the diameter and clearance ability of MLV was sensitive to single tPBM, which might explain how it affect the microglia states. Overall, the paper touched on several important questions in the field and provided phenotypic characterization for potential clinical interventions. Yet unfortunately, the logic of the study failed to provide convincing evidence to support major conclusions, especially on causal relationship between molecules, cells and phenotypes, in most figures only correlations are shown. The characterization of microglia is also relatively descriptive, lacking mechanistic insights. Below I listed the major points that I hope the author to clarify before considering for publication.

Major points:

1. The major topic for the study is to delineate the effect of transcranial photobiomodulation (tPBM) on diabetes, focusing on microglia and its related phenotypes like cerebrovascular injuries. However, throughout the work, the author did not provide any test on the outcome of diabetes and its relationship with either tPBM or microglia. Although microglia have been intensively reviewed to contribute to disease progression, it is difficult to judge how tPBM benefit microglia-related disease symptoms that are linked with diabetes. Using artificial-generated cerebral injury in Fig. 2 and 3 can only test the microglia motility, rather than to reflect any phenotypes that are related with diabetes. While the leakage of BBB might be an indication of diabetes-induced brain changes, it was not affected by tPBM treatment and might not related with microglia dysfunction. Therefore, all of the data failed to support the major conclusion raised by the author, which "tPBM can be an efficient method to prevent diabetic brain disorders". The effect of tPBM on microglia and on the diabetic disorders are logically separated and are not reconciled in the work. In other words, the author only showed tPBM+insulin could reverse microglia dysfunction (but see comments below this is also weak), but not supporting its contribution to disease.

2. The author only accessed the morphology and motility of microglia as an indication of their activation states, which was too descriptive as the morphology of microglia could be altered in response to diverse stimuli. The lack of other microglia characterization, including its proliferation, phagocytosis, secretion of cytokines and chemokines, and transcriptional changes, make it difficult to evaluate the changes of microglia following diabetes and after tPBM. Currently the field has generally recognized that single morphological description of microglia could not reflect its biology, therefore adding more systematic information of microglia to show their changes is needed.

3. The data in Fig. 4, which is the key for the study, clearly illustrate the tPBM alone did not affect the dysfunction of microglia, while the combination of insulin and tPBM together exhibited additive effects. However, the authors did not provide any mechanistic evidence on how such combination works. The reduction of IFN- γ might be one reason, but only correlation cannot provide causal relationship. How is IFN- γ level sensitive to the combination of insulin and tPBM, but not tPBM alone? Whether IFN signaling contribute to microglia dysfunction and diabetic phenotypes? Lack of causal studies cannot explain the biological phenotype especially in such a complex pathological situation. Actually such problem is generally applied to most of the data in the current study.

4. In Fig 5, the author applied NOS inhibitor L-NAME in DM background, trying to get the conclusion that the tPBM restored the MLV during DM that depends on the NOS pathway. However, the L-NAME alone (in non-DM mice) as the critical control is lacking, which makes it controversial to distinguish whether tPBM failed to rescue DM-related changes, or just simply cannot restore changes that are directly caused by L-NAME application. Also, the NOS pathway may only affect MLV diameter without having any contributing to the microglia dysfunction, IFN- γ level and DM phenotypes, at least without clear data supported.

Minor suggestions:

1. The authors keep using microglia activation to describe the changes of microglia during diabetes. Based on increasing understanding of microglia biology, it is now generally agreed that microglia are always active but in different states (Rosa C Paolicelli, et al, Neuron, 2022). I would suggest the author to use more precise way for the annotation of microglia, e.g. describing them with precise term like homeostatic status, responding to DM, etc.

2. The authors describe P2Y12 as a chemokine receptor. Please change it to the purinergic receptor as chemokine typically describe small proteins belongs to the cytokine family. P2Y12 senses ADP which belongs to purinergic transmitters.
3. The author did not provide individual data value in all the graphs, only showing the average points and error bar. With limited n number (only 4-5 in most experiments) and the statistics (ANOVA) used by the author, I would suggest to show all individual data points to better illustrate the variation of data. Also, with small n number and statistics conducted between two datasets, suggest to use Student t-test.
4. The tPBM was conducted with a fixed wavelength and duration. Please state why such parameter was chosen, or test other parameters and evaluate their effect on microglia and DM phenotypes.
5. The figure presentation could be further optimized. Following the text it jumped frequently from figure 3 to figure 1 and back to 3 again. Also, suggest to combine representative images with quantification in one figure, for example now figure 2 only show multiple raw images and difficult to get any conclusions without combining with figure 3.
6. The statistic for Figure 3 is too crowded. Can the author extract some features from the curve, e.g. duration from 5%-95%, time to reach 50% of max for quantification? It will make it easier to understand and conclusive.

Response to Reviewers

We greatly appreciate the great assistance with our manuscript (COMMSBIO-23-0993) and the constructive comments by the reviewers. We enclose the revised manuscript, which addresses all reviewers' comments point-by-point as detailed below. All changes in our manuscript are highlighted in yellow.

Sincerely,

Dan Zhu,

Ph.D., SPIE Fellow

Professor, Britton Chance Center for Biomedical Photonics, Huazhong University of Science and Technology

Deputy-Director, Wuhan National Laboratory for Optoelectronics

Secretary General & Vice-President, Biomedical Photonics Committee of Chinese Optical Society

Wuhan 430074, P.R. China

E-mail: dawnzh@mail.hust.edu.cn

Reviewer #1:

The current manuscript by Liu et al., is a rigorous and elegant study where the authors analyze the potential beneficial effect of PBM on microglial morphology and brain drainage system using an Alloxan-induced model of diabetes mellitus. The use of various methodologies and multidisciplinary approaches allow robustness to the conclusions.

The manuscript can be endorsed for publication in its current state by responding to the following minor comments

Comment 1: It is not clear why figure 5 does not include the groups DM-2W+I-1W and DM-2W+I&tPBM-1W as in the previous figures.

Response: The authors would like to express their sincere gratitude to the review for the great help in improving our paper and for the important remark. To investigate the effect of tPBM on the meningeal lymphatics of diabetic mice (revised Figure 6), we performed a single session of laser irradiation (the sequence of 17 min – irradiation, 5 min – pause, 61 min in total) on diabetic mice, instead of the 7 days tPBM course, and then analyzed the structural and functional changes in the meningeal lymphatic vessels. So, the groups DM-2W+I-1W and DM-2W+I&tPBM-1W were not included. However, according to the reviewer's suggestion, we also evaluated the changes of MLVs diameter after 7-day insulin and insulin+tPBM treatment (Supplementary Fig. 12).

Page 13, Lines 352-359: Furthermore, we also investigated the effects of 7-day course insulin/tPBM and insulin+tPBM on the diameter of MLVs in diabetic mice. Supplementary Fig. 12 showed that only insulin treatment could not increase the diameter of MLVs in diabetic mice (17.46 μm for DM+I vs. 16.39 μm for DM-3W, $p=0.727$), while both tPBM and tPBM combined with insulin treatment significantly increased the diameter of MVLs in diabetic mice (20.79 μm for DM+tPBM vs. 16.39 μm for DM-3W, $p<0.001$ and 21.51 μm for DM+I&tPBM vs. 16.39 μm for DM-3W, $p<0.001$). These results suggest that only tPBM has the ability to increase the diameter and clearance function of MLVs contributing to increase lymphatic function, which was also shown by us earlier (*Nat Commun* 14, 6104, 2023).

Supplementary Fig. 12 Effects of tPBM (7-day courses) on the diameter of the MLVs. a Fluorescence images of the basal MLVs (labeled with Lyve-1, green and Prox-1, yellow) and diameter distribution histogram in the tested groups. Scale bar: 50 μm . **b** Quantitative analysis of the diameter of the MLVs in the tested groups. Data were presented as mean \pm standard deviation (n=6 mice), statistical significance was assessed using one-way ANOVA (n.s., *, **, and *** indicate not significant, $p < 0.05$, $p < 0.01$ and $p < 0.001$, respectively).

Comment 2: In addition, representative fluorescent images of the basal MLVs (labeled with Lyve-1, green and Prox-1, yellow) and diameter distribution histogram in the control group with only L-NAME treatment should be included.

Response: We highly appreciate the reviewer's helpful comments. And we added the representative fluorescence images of the basal MLVs (labeled with Lyve-1, green and Prox-1, yellow) and diameter distribution histogram in the control group with only L-NAME treatment in the Figure 6.

Page 13, Lines 360-372: The mechanisms of the tPBM effects on the MLVs remain unclear. There is evidence that it might be via tPBM-mediated stimulation of nitric oxide (NO) production in the lymphatic endothelium⁴⁰⁻⁴³. To verify the role of NO in the tPBM-mediated stimulation of the MLVs, we investigated the effect of blockage of NO-synthases using L-NAME on the tPBM-mediated changes in the diameter of the MLVs in DM-3W mice. First, we inhibited the release of NO in normal mice with L-NAME and found that the diameter of the MLVs in normal mice was significantly reduced (19.69 μm in CTR group vs. 16.73 μm in CTR+L-NAME group, $p = 0.017$. Fig. 6c and d). Then, we also treated DM-3W mice with L-NAME, but the diameter of MLVs in diabetic mice was not changed (16.39 μm in DM-3W group vs. 16.27 μm in DM+L-NAME group, $p = 0.999$. Fig. 6c and d). However, our results also showed that there were no changes in the diameter of the MLVs after the single course of tPBM in the diabetic mice treated with L-NAME (17.62 μm for

DM+L-NAME+tPBM vs. 16.27 μm for DM+L-NAME, $p=0.613$. Fig. 6c and d), i.e. the NO blockade completely suppressed tPBM-mediated modulation of the MLVs.

Fig. 6 Effects of tPBM (single application) on lymphatic removal of EBD and on the diameter of the MLVs. **a** Fluorescence images of EBD clearance from the right lateral ventricle into the dcLNs in the tested groups. Scale bar: 1 mm. **b** Quantitative analysis of fluorescence intensity of EBD in the dcLNs of the tested groups. **c** Representative fluorescent images of the basal MLVs (labeled with Lyve-1, green and Prox-1, yellow) and diameter distribution histogram in the tested groups. Scale bar: 50 μm . **d** Quantitative analysis of the diameter of the MLVs in the tested groups. Data were presented as mean \pm standard deviation ($n=6$ mice), statistical significance was assessed using one-way ANOVA (n.s., *, **, and *** indicate not significant, $p<0.05$, $p<0.01$ and $p<0.001$, respectively).

Comment 3: It is not clear in the discussion, why PBM decreases inflammation (relative expression of IFN) but the BBB remains permeable and does not reverse the changes in the expression of ZO1/Occludin as in the DM-3W group.

Response: Thanks very much for your comment. We have added a discussion of this issue.

Page 14, Lines 416-424: Diabetes enhances the BBB leakage due to tight junction protein loss, but hyperglycemia and inflammation are not the only damaging factor responsible for the BBB leakage in diabetes³⁵. A large number of studies have shown that diabetes-related hypertension, increased oxidative stress, hyperlipidemia, and insulin resistance contribute to the development of BBB dysfunction⁵¹⁻⁵³. Although the tPBM treatment could effectively decrease inflammation in the brain tissue, it needs further investigation whether tPBM has the ability to reverse these changes, increase the expression of tight junction proteins and restore the permeability of BBB. This might explain why tPBM reduced the level of inflammatory factors, but failed to increase tight junction proteins

ZO-1/Occludin expression.

Reviewer #2:

The study is original and presents interesting results to better understand the influence of tPBM and the cerebral mechanisms of action involved, in the context of the treatment of diabetes. The article is well written, researched and accurate.

Finally, the discussion is original and well constructed.

Comment 1: It would be interesting to specify why this diabetic mouse model was chosen, by comparing with the existing one in the literature.

Response: We would like to express our sincere gratitude to the review for the great help in improving our paper and for the important remark. Commonly used animal models of type I diabetes include pancreatic injury model, transgenic model, spontaneous animal model, and chemical substance-induced model (*Can J Diabetes*. 2013, 37 (4):269-276; *Nat Rev Endocrinol*. 2018, 14 (3):140-162). The pancreatic injury model is targeted to damage the islets without affecting other tissues and organisms, but it has a high mortality rate due to hypoglycemia caused by pancreatic α cell damage (*Endocrinology*. 2011, 152 (7):2580-2588). The spontaneous animal model has a stable genetic background and high consistency of experimental data, but it is costly (*Biochem Pharmacol*. 2016, 99:1-10). The transgenic model allows *in vivo* knockout of diabetes-related polygenes and the study of single-gene function, but the techniques and the screening process are complex, the breeding and feeding requirements are highly demanding (*J Diabetes Res*. 2016, 2016:9051426). The chemical substance-induced model, compared with the pancreatic injury model, selectively damage β cells and can effectively reduce mortality; compared with the transgenic and spontaneous models, it has the advantages of simplicity in modeling, short modeling cycle, economical and easy to use for a large number of animals. Therefore, the chemical substance-induced model is widely used in the study of diabetes (*Br J Pharmacol*. 2012, 166(3):877-894; *J Pharm Pharmacol*. 2010, 62(1):1-23). In this study, we chose a commonly used chemical substance-induced diabetic mouse model (*Medicina*. 2017, 53(6):365-374; *Lab Anim Res*. 2021,37(1):23; *Diabetes* 2019, 68(S1):1748-P; *Diabetes*. 2018, 67(8):1576-1588).

Comment 2: Could you specify for the choice of the wavelength of the tPBM, and the time slot for tPBM treatment?

Response: We highly appreciate the reviewer's comments. In our previous works, we have demonstrated that the near-infrared wavelengths (1267 nm) laser, and 7-day courses (17 min-irradiation, 5 min-pause, 3 cycles) tPBM have the ability to stimulate the meningeal lymphatic vessels. And this treatment can be therapeutically effective in several disease models. For example, reducing the A β burden in Alzheimer's disease mice (*Biomed Opt Express* 2019, 10(8):4003-4017), accelerating red blood cells clearance in mice with intraventricular hemorrhage (*Nat Commun* 14, 6104, 2023), and consequently improving the behaviours of the mice. Therefore, the same wavelength and the time slot for tPBM treatment was used in this work.

Comment 3: Could you also specify the local increase in temperature, at the level of the tissue, if there is one?

Response: We highly appreciate the reviewer's helpful advice. We added additional experiments to measure changes in mouse cortical temperature during tPBM (Supplementary Fig. 9).

Page 10, Lines 273-277: Besides, to demonstrate the safety of tPBM, we measured changes in mouse cortical temperature during one course of tPBM in the CTR group. The results showed that the temperature on the cortex surface was increased only about 0.4 °C during tPBM (Supplementary Fig. 9), which suggested that tPBM did not have a significant thermal effect.

Supplementary Fig. 9 Changes in cortex temperature induced by tPBM. Data were presented as mean \pm standard deviation (n=5 mice).

Comment 4: Note that the method of anesthesia is also likely to affect the functioning of the BBB.

Response: We highly appreciate the reviewer's helpful advice. In our *in vivo* experiments to evaluate the permeability of BBB, all the mice in different groups were anesthesia in the same way. Diabetic mice showed increased permeability of the BBB, however, the BBB remained integrity in healthy mice. Besides, for quantitative assessment of changes in BBB permeability, brain tissue was analyzed *ex vivo* rather than under anesthesia. Thus, these results indicated that changes in BBB function were not induced by anesthesia but by diabetes.

Comment 5: Regarding the results. The study is complete; nevertheless the presentation of the graphs and their naming is not implicit easily. A summary figure of the groups and durations of treatment, and comparisons made will facilitate the understanding of the article.

Response: We highly appreciate the reviewer's constructive advice. We have added a summary figure of the tested groups and durations of treatment (Supplementary Fig. 2) in the revised manuscript. The comparisons made were described in the manuscript and presented in each figure.

Page 17, Lines 537-540: The course of tPBM for 7 days was performed in the following groups: 1) CTR (CTR+tPBM); 2) DM-2W (DM+tPBM); 2) DM+I (DM+I&tPBM). We used the group of DM-3W as a comparison because the insulin/tPBM treatment was performed during 7 days in the DM-2W and when we analyzed the microglial morphology and function, these mice were already DM-3W.

Supplementary Fig. 2 The summary figure of the tested groups and durations of treatment.

Comment 6: The number of 4 animals per group is low, but nevertheless allows us to observe significance, a sign that the effect of the treatments is indeed present. Nevertheless, concerning the statistical analysis of the data, it should be indicated what type of ANOVA is carried out, and whether the measurements are repeated or not. The mice until the time of treatment should be n=12 in number, in a homogeneous group (box plot) and then divided into 3 groups of 4: DM-3W, DM-2W+I-1W, DM-2W+ I-1W+tPBM.

Response: We highly appreciate the reviewer's constructive advice. The statistical analysis was described in the legends of each figure. And we also provided a description of the detailed statistical methodology in the revised manuscript.

Pages 21-22, Lines 725-738: Statistical analysis was performed using the SPSS (IBM) software. Sample sizes (n) are presented in the figure legends, with animals randomly assigned to groups for experiments. And the n value was defined as the number of independent animals. Data are presented as the mean \pm standard deviation. The Shapiro-Wilk test, a method for small sample sizes, was used to assess the normality of data distribution in each experiment. The heterogeneity of variance was evaluated using the Levene test, a stable method for both normally and nonnormally distributed data. The significance of differences between means was evaluated by unpaired Student's t test (normality distribution, variance homogeneity) for two independent group comparisons. And one-way ANOVA with Turkey's multiple-comparison test (variance homogeneity) or Dunnett's T3 multiple-comparison test (variance non-homogeneity) was used for comparisons of more than two groups. Repeated measurements ANOVA was used when analyzing the changes with the development of diabetes, and non-repeated measurements ANOVA was used when analyzing the changes induced by insulin/tPBM and insulin+tPBM treatment. All p values < 0.05 were considered statistically significant (* p < 0.05, ** p < 0.01, and *** p < 0.001).

Besides, we also revised the description of the grouping of mice.

Page 17, Lines 541-543: Actually, the mice until the time of treatment were n=16 in number, in a homogeneous group and then divided into 4 groups of 4: DM-3W, DM+I, DM+tPBM, DM+I&tPBM.

Reviewer #3:

In the current manuscript, Liu et al. aimed to search for new methods to restore the dysfunction of microglia during diabetes and to improve the pathology. They first established the diabetes model in mice through alloxan injection, and described the changes in morphology and motility of microglia that followed the progression of disease. Next, using transcranial photobiomodulation with 1267nm near infrared light, they demonstrated the beneficial effect on reversing microglia dysfunction, which correlated with the changes of P2Y12R level. Finally, they checked the brain lymphatic system after tPBM and found the diameter and clearance ability of MLV was sensitive to single tPBM, which might explain how it affect the microglia states. Overall, the paper touched on several important questions in the field and provided phenotypic characterization for potential clinical interventions. Yet unfortunately, the logic of the study failed to provide convincing evidence to support major conclusions, especially on causal relationship between molecules, cells and phenotypes, in most figures only correlations are shown. The characterization of microglia is also relatively descriptive, lacking mechanistic insights. Below I listed the major points that I hope the author to clarify before considering for publication.

Major points:

Comment 1: The major topic for the study is to delineate the effect of transcranial photobiomodulation (tPBM) on diabetes, focusing on microglia and its related phenotypes like cerebrovascular injuries. However, throughout the work, the author did not provide any test on the outcome of diabetes and its relationship with either tPBM or microglia. Although microglia have been intensively reviewed to contribute to disease progression, it is difficult to judge how tPBM benefit microglia-related disease symptoms that are linked with diabetes. Using artificial-generated cerebral injury in Fig. 2 and 3 can only test the microglia motility, rather than to reflect any phenotypes that are related with diabetes. While the leakage of BBB might be an indication of diabetes-induced brain changes, it was not affected by tPBM treatment and might not related with microglia dysfunction. Therefore, all of the data failed to support the major conclusion raised by the author, which “tPBM can be an efficient method to prevent diabetic brain disorders”. The effect of tPBM on microglia and on the diabetic disorders are logically separated and are not reconciled in the work. In other words, the author only showed tPBM+insulin could reverse microglia dysfunction (but see comments below this is also weak), but not supporting its contribution to disease.

Response: We'd like to express our gratitude for the constructive advice. We have added some studies on the effects of insulin/tPBM treatment on the physiology of diabetic mice.

Page 10, Lines 280-291: In addition to microglia, we also investigated the effects of insulin/tPBM on the physiology of diabetic mice. The energy expenditure, respiratory exchange ratio (RER) and locomotor activity were analyzed to investigate the changes in energy balance during the development of DM in mice (Supplementary Fig. 10). The results revealed that energy expenditure and locomotor activity were significantly reduced that was associated with a decrease in RER in DM-3W mice compared with the CTR group. The combined therapy with insulin and tPBM improved physical activity and changed RER in DM mice. Indeed, energy expenditure, locomotor activity and RER were approximately identical to those in the control. The separate therapy with insulin or tPBM had not full therapeutic effects in DM mice. So, insulin increased RER in DM-3W mice, while energy expenditure and locomotor activity did not statistically change from the DM-

3W group. The therapy with tPBM did not influence on RER and slightly changed energy expenditure and locomotor activity that was not statistically differ from the DM-3W group.

Supplementary Fig. 10 The analysis of energy and locomotor activity changes in mice. **a** The system for automatic recording of behavioral and metabolic parameters in mice (PhenoMaster TSE Systems). **b-d** Quantitative analysis of changes in energy expenditure (**b**), respiratory exchange ratio (RER) (**c**) and locomotor activity (**d**) in mice. Data were presented as mean \pm standard deviation (n=5 mice).

Pages 15-16, Lines 442-485: Our results also revealed significant changes in energy balance in DM-3W compared with the control. Indeed, DM was associated with a decrease in physical activity and energy expenditure that was accompanied by a reduction of RER indicating that fat was the predominant fuel source for DM mice. The combined therapy with insulin and tPBM significantly improved metabolism and physical activity in DM mice that it was established by the restoration of the indicated parameters. The separate therapy with insulin or tPBM had not result in full recovery of metabolism and locomotor activity in DM mice suggesting that insulin and tPBM enhance each other's therapeutic effects leading to a significant improvement of metabolism and energy balance. Our findings are consistent with other results suggesting that tPBM ameliorates curing diabetic complications, such as diabetic foot⁶², diabetic periodontitis⁶³, and diabetic retinopathy⁶⁴. Recent studies indicate that tPBM improves insulin sensitivity in high-fat diet-induced mice^{65, 66}. TPBM also improves glucose and lipid metabolism disorders in white adipose tissues^{65, 66} and 3T3-L1 adipocytes⁶⁷. The tPBM-mediated activation of cytochrome c oxidase (CcO)-mediated protein kinase B can be one of mechanisms of therapeutic effects of tPBM for DM⁶⁸. TPBM accelerates mitochondrial ATP and ROS generation by increasing mitochondrial respiratory chain CcO activity. TPBM-related increase in ATP generation might elevate lipid consumption and attenuate fat deposition in skeletal muscle, whereas the transient ROS production induced by tPBM can promote activation of the PTEN/AKT/GLUT4 and PTEN/AKT/GSK-3 β /GS pathways, increasing glucose absorption and glycogen accumulation in skeletal muscle.

How tPBM can improve the outcome of DM remains poorly understood. Emerging evidence indicates the important homeostatic and pathophysiological roles of MLVs in the progression of various cerebral small vessel diseases, including DM⁶⁹⁻⁷². Indeed, genetic and pharmacological disruption of MLVs results in less drainage of CSF and ISF to dLN^{34, 73}. Such disruption also results in cognitive impairment and behavioral alterations due to dysfunctions of brain's waste removal system leading to accumulation of toxins and metabolites in the CNS⁷³. The dysfunction of the MLVs might provide an important contribution to age-related cognitive decline and neurodegenerative disease^{74, 75}. The MLVs affect microglia responses⁷³. In the normal condition, MLVs naturally contain a substantial pool of immune cells, such as T cells, macrophages, dendritic cells, mast cells, neutrophils, monocytes, macrophages, and B cells⁷⁶⁻⁷⁸. Under brain pathology, these immune cells infiltrate the brain parenchyma from CNS borders through the direct vascular-like channels and further triggered a series of immune responses of neurons and microglia⁷⁹. For instance, T cells in the meninges modulate behavior and cognition by releasing cytokines in mice⁸⁰. Emerging evidence has highlighted the capacity of hematogenous cells in skull bone marrow to enter the meninges via ossified vascular channels and maintain immune homeostasis in the CNS. CNS-adjacent skull bone marrow comprises hematopoietic niches that can sense CNS injury and supply specialized immune cells to fine-tune inflammatory responses. From this aspect, the brain drainage system, including MLVs, is considered a messenger regulating neuroinflammation and neuroimmune^{69, 81}. Interestingly, the stimulation of meningeal lymph angiogenesis via administration of the vascular endothelial growth factor can improve the drainage of macromolecules to the cervical lymph nodes leading to an increase in resistance to brain diseases^{73, 82-85}. Thus, tPBM-mediated augmentation of the MLVs functions, that we showed in our previous works also²⁰, can be one of mechanisms of improvement of therapeutic effects of insulin, including microglia reactivity as a part of tPBM-related regulation of neuroimmune providing restoration of homeostasis in the CNS thereby improving recovery of energy balance in DM+I mice.

Comment 2: The author only accessed the morphology and motility of microglia as an indication of their activation states, which was too descriptive as the morphology of microglia could be altered in response to diverse stimuli. The lack of other microglia characterization, including its proliferation, phagocytosis, secretion of cytokines and chemokines, and transcriptional changes, make it difficult to evaluate the changes of microglia following diabetes and after tPBM. Currently the field has generally recognized that single morphological description of microglia could not reflect its biology, therefore adding more systematic information of microglia to show their changes is needed.

Response: We'd like to express our gratitude for the constructive advice. We have added more systematic information of microglia (Supplementary Fig. 5) to show their changes in the revised manuscript.

Page 8, Lines 223-238: To systematically analyze the effects of insulin/tPBM treatment on microglia in diabetic mice, we also evaluated the proliferative properties of microglia and the secretion of cytokines (Supplementary Fig. 5). Here, we used anti-DAPI antibody to stain the nuclear, and the anti-Mki67 antibody was used to stain the proliferative cells in the brain and analyzed the percentages of the proliferative microglia (KI67⁺ microglia) in different groups. As shown in Supplementary Fig. 5a, there were no KI67⁺ microglia in all the tested groups, which meant the proliferation of microglia was not changed. In response to changes in the

microenvironment of the brain, microglia secrete a variety of cytokines. Herein, we also assessed the changes of the level of several cytokines' mRNA expression. The results in Supplementary Fig. 5b-e showed that mRNA expression of pro-inflammation cytokines (CD86 and TNF- α) was notably increased, while the mRNA expression of anti-inflammation cytokines (CD206 and TGF- β) in the DM-3W group, both insulin or tPBM alone treatment could reduce the mRNA expression of pro-inflammation cytokines and increase the mRNA expression of anti-inflammation cytokines. These effects further enhanced by insulin combined with tPBM treatment. These results suggested that more microglia converted towards an anti-inflammatory phenotype after insulin combined with tPBM treatment, to protect against diabetes-induced tissue damage.

Supplementary Fig. 5 The changes of microglial characterization. **a** Changes of microglial proliferation in the tested groups. **b-e** Changes of the expression of mRNA level of different microglia cytokines CD86 (**b**), CD206 (**c**), TNF- α (**d**), TGF- β (**e**) in the tested groups. Data were presented as mean \pm standard deviation (n=3 mice).

Comment 3: The data in Fig. 4, which is the key for the study, clearly illustrate the tPBM alone did not affect the dysfunction of microglia, while the combination of insulin and tPBM together

exhibited addictive effects. However, the authors did not provide any mechanistic evidence on how such combination works. The reduction of IFN- γ might be one reason, but only correlation cannot provide causal relationship. How is IFN- γ level sensitive to the combination of insulin and tPBM, but not tPBM alone? Whether IFN signaling contribute to microglia dysfunction and diabetic phenotypes? Lack of causal studies cannot explain the biological phenotype especially in such a complex pathological situation. Actually, such problem is generally applied to most of the data in the current study.

Response: We'd like to express our gratitude for the constructive advice. In the previous version of the manuscript, the data showed that insulin alone have some therapeutic effects on the dysfunction of microglia induced by diabetic, and insulin+tPBM further enhanced the therapeutic efficacy. But we didn't show the effect of tPBM alone treatment. In the revised manuscript, to better understand the effects of insulin and tPBM, we evaluated the functional changes of microglia in diabetic mice treated with insulin/tPBM and insulin+tPBM (Fig. 3 and Fig. 4). Either insulin or light treatment alone partially improved microglial function, and the combination of insulin and tPBM together enhanced each other's effectiveness.

Fig. 3 Changes of microglial density and morphology induced by diabetes and insulin/tPBM treatment. **a** Representative images of density of cortical microglia in the tested groups. Scale bar: 100 μm . **b** Representative images of morphological changes of microglia in the tested groups. Scale bar: 25 μm . **c** Three-dimensional reconstruction of microglia cells in the tested groups. Scale bar:

25 μm . **d** Enlarged views of microglia in the green dashed box in (c). Scale bar: 10 μm . **e-h** Quantitative analysis of cortical microglial density (e), soma volume (f), soma sphericity (g), and average percentage area of microglia (h) in the tested groups. Data were presented as mean \pm standard deviation ($n=5$ mice in each group), statistical significance was assessed using one-way ANOVA (n.s., *, **, and *** indicate not significant, $p<0.05$, $p<0.01$ and $p<0.001$, respectively).

Fig. 4 Quantification analysis the functional changes of microglia in response to vascular injury in diabetic and insulin/tPBM treatment mice. **a** Representative two-photon microscopic images of the time-depended changes in the microglia (green) around the photo-damaged area (white arrow) of the cerebral vessels filled by the RB (red) in the tested groups. Scale bar: 25 μm . **b** The changes of polygon area bound by microglial branches closest to the injury in the tested groups.

c Quantitative analysis of the time taken for the area of the polygon to be reduced to a minimum. **d** Quantification of the maximum percentage of microglial branches accumulation at different distances from the injury in the tested groups. **e** Quantification of the percentage of microglial branches accumulation at different distances 60 min after the injury in the tested groups. Data were presented as mean \pm standard deviation (n=4 mice), statistical significance was assessed using one-way ANOVA (n.s., *, **, and *** indicate not significant, $p < 0.05$, $p < 0.01$ and $p < 0.001$, respectively).

In previous research, Taylor et al. found that microglial response to vascular injury was impaired in diabetic mice. And they also demonstrated that microglial dysfunction is caused by chemotaxis impairment. Notably, blocking IFN- γ by IFN- γ -neutralizing antibody in diabetic mice restored microglial process polarity toward the microbleeds and reduced secondary leakage (*J Neurosci.* 2018, 38(40):8707-8722). These findings indicate that hyperglycemia-induced brain inflammation impairs the microglial response to vascular injury through IFN- γ signaling. How do increases in IFN- γ impair microglial function? Taylor et al. suggested that microglial dysfunction was caused by chemotaxis impairment. The P2Y₁₂R, which has an important role in mediating microglial chemotaxis to BBB repair (*Nat Neurosci.* 2006, 9(12):1512-1519), was downregulated in diabetic mice. Notably, both DEX and anti-IFN- γ treatments rescued P2Y₁₂R expression levels (*J Neurosci.* 2018, 38(40):8707-8722). Together, these results indicate that microglial recognition and migration to the damaged vessel is impaired in response to insulin depletion, and that P2Y₁₂R and IFN- γ are key regulators of microglial dysfunction (*J Neurosci.* 2019, 39(24):4632-4635).

The inability to repair injured vessels also aggravates the disruptive effect that permeable blood vessels have on local synaptic structure and function, thereby compromising synaptic plasticity and memory (*PLoS Biol.* 2007, 5(5):e119; *PLoS One.* 2013, 8(5):e65663; *J Neurosci.* 2015, 35(13):5128-5143; *Neurobiol Dis.* 2015, 78:1-11). Indeed, vascular damage has been associated with cognitive impairment and risk of dementia (*Stroke*, 2007, 38(6):1949-1951; *Alzheimer Dis Assoc Disord.* 2014, 28(2):106-112; *J Neurol Sci.* 2016, 368:195-202). In addition, patients with diabetes have a higher risk of the development of dementia, and diabetes is more prevalent among patients with dementia (*Diabetologia.* 1996, 39(11):1392-1377; *Lancet Neurol.* 2006, 5(1):64-74). Therefore, vascular complications might be a link in the relationship between these disorders. Alzheimer's disease (AD) and vascular dementia (VaD) are the most prevalent forms of dementia among patients with diabetes (*Lancet Neurol.* 2006, 5(1):64-74; *Nat Rev Endocrinol.* 2018, 14(10):591-604). Notably, the presence of cerebral microbleeds also correlates with decreased amyloid- β peptide levels in CSF, as in patients with AD (*J Alzheimers Dis.* 2017, 55(3):905-913; *Neuroradiol J.* 2017, 30(4):330-335). In addition, IFN- γ levels are increased in the sera of patients with VaD (*Neurobiol Aging.* 2015, 36(9):2597-2606) and AD (*J Neurol Neurosurg Psychiatry.* 2017, 88(10):876-882). Moreover, a study in a small cohort showed that peripheral IFN- γ levels are higher in patients with mild to severe AD, compared with control subjects (*J Interferon Cytokine Res.* 2014, 34(11):839-847). Transgenic mouse models of AD also display increased levels of IFN- γ , and treatment with an anti IFN- γ antibody attenuates microglial activation and memory deficits (*J Immunol.* 2013, 190(5):2241-2251). Finally, recent work indicated that cerebrovascular damage can be an early biomarker of dementia in patients with AD (*Nat Med.* 2019, 25(2):270-276). Together, these results suggest that increased IFN- γ levels observed in patients with diabetes might increase the risk of dementia. Notably, inflammation and vascular damage have been widely associated with neurodegenerative disorders, such as VaD and AD. Data from the study by Taylor et al. (*J Neurosci.*

2018, 38(40):8707-8722), along with previous findings on vessel damage and dementia, encourage future studies on IFN- γ as a therapeutic target for both diabetes and dementia.

Comment 4: In Fig 5, the author applied NOS inhibitor L-NAME in DM background, trying to get the conclusion that the tPBM restored the MLV during DM that depends on the NOS pathway. However, the L-NAME alone (in non-DM mice) as the critical control is lacking, which makes it controversial to distinguish whether tPBM failed to rescue DM-related changes, or just simply cannot restore changes that are directly caused by L-NAME application. Also, the NOS pathway may only affect MLV diameter without having any contributing to the microglia dysfunction, IFN- γ level and DM phenotypes, at least without clear data supported.

Response: We highly appreciate the reviewer's helpful comments. And we added the representative fluorescent images of the basal MLVs (labeled with Lyve-1, green and Prox-1, yellow) and diameter distribution histogram in the control group with only L-NAME treatment in the Figure 6.

Page 13, Lines 362-372: To verify the role of NO in the tPBM-mediated stimulation of the MLVs, we investigated the effect of blockage of NO-synthases using L-NAME on the tPBM-mediated changes in the diameter of the MLVs in DM-3W mice. First, we inhibited the release of NO in normal mice with L-NAME and found that the diameter of the MLVs in normal mice was significantly reduced (19.69 μm in CTR group vs. 16.73 μm in CTR+L-NAME group, $p=0.017$. Fig. 6c and d). Then, we also treated DM-3W mice with L-NAME, but the diameter of MLVs in diabetic mice was not changed (16.39 μm in DM-3W group vs. 16.27 μm in DM+L-NAME group, $p=0.999$. Fig. 6c and d). However, our results also showed that there were no changes in the diameter of the MLVs after the single course of tPBM in the diabetic mice treated with L-NAME (17.62 μm for DM+L-NAME+tPBM vs. 16.27 μm for DM+L-NAME, $p=0.613$. Fig. 6c and d), i.e. the NO blockade completely suppressed tPBM-mediated modulation of the MLVs.

As L-NAME treatment didn't cause changes in the diameter of MLVs in diabetic mice, while it completely suppressed tPBM-mediated modulation of the MLVs. These results showed tPBM failed to rescue DM-related changes after L-NAME application, and indicated that tPBM restored the MLV during DM that depends on the NOS pathway.

In recent years, with the discovery of the MLVs, many studies have shown that the MLVs play an important role in the clearance of various substances from the brain, the substances in the brain could be removed through the MLVs to the deep cervical lymph nodes (dCLNs). In our recent work, we found that the same way tPBM treatment could accelerate red blood cells clearance in mice with intraventricular hemorrhage. And we demonstrated that tPBM effects on the lymphatic red blood cells clearance can be realized mainly through NO-mediated increase in MLVs diameter, as the blockage of NO-synthases using L-NAME suppressed tPBM-induced dilatation of MLVs and significantly reduced the number of RBCs clearance (*Nat Commun* 14, 6104, 2023). In this work, we also found that the tPBM-mediated increase in MLVs diameter was significantly inhibited by L-NAME in diabetic mice. Hence, we expected that lymphatic clearance of inflammatory factors (IFN- γ) will be inhibited, which in turn affect microglia function and diabetic phenotypes.

Minor suggestions:

Comment 1: The authors keep using microglia activation to describe the changes of microglia during diabetes. Based on increasing understanding of microglia biology, it is now generally agreed that microglia are always active but in different states (Rosa C Paolicelli, et al, *Neuron*, 2022). I

would suggest the author to use more precise way for the annotation of microglia, e.g. describing them with precise term like homeostatic status, responding to DM, etc.

Response: We highly appreciate the reviewer's constructive suggestion. We have revised the description of microglial states.

Page 2, Lines 83-86: However, the microglial morphology has altered significantly, suggesting microglia responding to DM. In fact, microglia in the CTR group were in a homeostatic status, with small ellipsoidal bodies and elongated branches.

Page 3, Lines 111-112: Thus, these results clearly demonstrate the DM-mediated changes of microglial morphology, suggesting microglia responding to DM in diabetic mouse was aggravated with the DM progression.

Comment 2: The authors describe P2Y₁₂ as a chemokine receptor. Please change it to the purinergic receptor as chemokine typically describe small proteins belongs to the cytokine family. P2Y₁₂ senses ADP which belongs to purinergic transmitters.

Response: We highly appreciate the reviewer's constructive suggestion. We have revised this mistake, and described P2Y₁₂ as a "purinergic receptor" in the revised manuscript.

Comment 3: The author did not provide individual data value in all the graphs, only showing the average points and error bar. With limited n number (only 4-5 in most experiments) and the statistics (ANOVA) used by the author, I would suggest to show all individual data points to better illustrate the variation of data. Also, with small n number and statistics conducted between two datasets, suggest to use Student t-test.

Response: We highly appreciate the reviewer's constructive suggestion. We added all individual data points in each figure in the revised manuscript.

The significance of differences between means was evaluated by unpaired Student's t test (normality distribution, variance homogeneity) for two independent group comparisons. And one-way ANOVA with Turkey's multiple-comparison test (variance homogeneity) or Dunnett's T3 multiple-comparison test (variance non-homogeneity) was used for comparisons of more than two groups.

Comment 4: The tPBM was conducted with a fixed wavelength and duration. Please state why such parameter was chosen, or test other parameters and evaluate their effect on microglia and DM phenotypes.

Response: We highly appreciate the reviewer's comments. In our previous works, we have demonstrated that the near-infrared wavelengths (1267 nm) laser, and 7-day courses (17 min-irradiation, 5 min-pause, 3 cycles) tPBM have the ability to stimulate the meningeal lymphatic vessels. And this treatment can be therapeutically effective in several disease models. For example, reducing the A β burden in Alzheimer's disease mice (*Biomed Opt Express* 2019, 10(8):4003-4017), accelerating red blood cells clearance in mice with intraventricular hemorrhage (*Nat Commun* 14, 6104, 2023), and consequently improving the behaviours of the mice. Therefore, the same wavelength and the time slot for tPBM treatment was used in this work.

Comment 5: The figure presentation could be further optimized. Following the text it jumped frequently from figure 3 to figure 1 and back to 3 again. Also, suggest to combine representative images with quantification in one figure, for example now figure 2 only show multiple raw images and difficult to get any conclusions without combining with figure 3.

Response: We highly appreciate the reviewer's constructive suggestion. We have optimized the figure presentation, and combined representative images with quantification in one figure in the revised manuscript.

Comment 6: The statistic for Figure 3 is too crowded. Can the author extract some features from the curve, e.g. duration from 5%-95%, time to reach 50% of max for quantification? It will make it easier to understand and conclusive.

Response: We highly appreciate the reviewer's constructive suggestion. Two parameters were extracted from the curve for quantification.

Page 6, Lines 172-175: In order to quantitative analysis the accumulation of microglial branches, we calculated the maximum percentage of microglial branches (Fig. 2g) and the percentage of microglial branches 60 min after the injury (Fig. 2h) in different distances.

The results were shown in Fig. 2h, g and Fig. 4d, e.

Fig. 2 The changes in the microglial reactivity to photoablation of the cerebral vessels in healthy and diabetic mice. **a** The design of study of the microglial responses to photoablation of the cerebral vessels. **b** Representative two-photon microscopic images of the time-depended changes in the microglia (green) around the photo-damaged area (white arrow) of the cerebral vessels filled by the RB (red) in the tested groups. Scale bar: 25 μm . **c** Schematic illustration of the extension velocity of the microglial branches to the photo-damaged area. The green radial lines divide the image into 36 sectors. The vertices of red polygon correspond to the microglial branches closest to injury in each sector. **d** Schematic illustration of the percentage of microglial branches accumulation within the photo-damaged area presented as circles with a spaced 5 μm . The center of the laser-induced bleeding (0-10 μm) was omitted due to the presence of autofluorescence signal. **e** The changes of polygon area bound by microglial branches closest to the injury in the tested groups. **f** Quantitative analysis of the time taken for the area of the polygon to be reduced to a minimum. **g** Quantification of the maximum percentage of microglial branches accumulation at different distances from the injury in the tested groups. **h** Quantification of the percentage of microglial branches accumulation at different distances 60 min after the injury in the tested groups. Data were

presented as mean \pm standard deviation (n=4 mice), statistical significance was assessed using one-way ANOVA (n.s., *, **, and *** indicate not significant, $p < 0.05$, $p < 0.01$ and $p < 0.001$, respectively).

Fig. 4 Quantification analysis the functional changes of microglia in response to vascular injury in diabetic and insulin/tPBM treatment mice. **a** Representative two-photon microscopic images of the time-depended changes in the microglia (green) around the photo-damaged area (white arrow) of the cerebral vessels filled by the RB (red) in the tested groups. Scale bar: 25 μ m. **b** The changes of polygon area bound by microglial branches closest to the injury in the tested groups. **c** Quantitative analysis of the time taken for the area of the polygon to be reduced to a minimum. **d** Quantification of the maximum percentage of microglial branches accumulation at different

distances from the injury in the tested groups. e Quantification of the percentage of microglial branches accumulation at different distances 60 min after the injury in the tested groups. Data were presented as mean \pm standard deviation (n=4 mice), statistical significance was assessed using one-way ANOVA (n.s., *, **, and *** indicate not significant, $p < 0.05$, $p < 0.01$ and $p < 0.001$, respectively).

REVIEWERS' COMMENTS:

Reviewer #1 (Remarks to the Author):

I endorse the manuscript for publication, since the authors satisfactorily responded to the comments made in the first part of the evaluation. Therefore, I consider that the manuscript has sufficient quality to be published in the journal.

Reviewer #2 (Remarks to the Author):

I would like to thank authors for answering all remarks and questions.

Reviewer #3 (Remarks to the Author):

In the revised manuscript by Liu et al., the authors have seriously taken my concerns into consideration, and performed additional experiments and analysis to support their main claims. Most of my questions are logically and evidently answered by the authors with new data, as well as citing related literatures in the field to support their conclusion. Also, updates that recognize the diversity of microglia were done. There only leave one point that I feel the author still did not address, which is listed below.

In my previous comment 1, I suggested the author to perform more causal studies rather than only reporting the correlation between phenotypes. While some part of the study it may be difficult to narrow down to a single target, and it is acceptable that the author provided one possible mechanism, like the IFN-gamma related part and the nNOS part, the major conclusion that touch microglia dysfunction should be causally revisited.

Based on the data in the first several figures, the author concluded in the title and abstract on their key finding, that reversing microglia dysfunction is the major reason (or at least one of the reason) for tPBM's effect on diabetes. Logically, this can be written as tPBM \diamond microglia \diamond diabetes. I understand that microglia may not be the only reason, but current data did not support their importance, rather only provided correlation. The logic presented in the figure can be summarized (after adding diabetes phenotype) as tPBM \diamond microglia, and tPBM \diamond diabetes, which are parallel. This is what I comment as correlation in my previous question. In another word, tPBM induced microglia changes may not even have any contribution to the tPBM induced diabetes improvement, although I agree it is highly possible given microglia are so important and went wrong during diseases. One possible experiment is to use loss of function on microglia, to see whether tPBM induced improvement on diabetes is affected. This could be achieved by using drugs to eliminate microglia, or to block P2Y12 receptors as the author suggested about their importance. Without such kind of experiments the key conclusion from the author is not well supported.

Response to Reviewer

We highly appreciate the great assistance with our manuscript (COMMSBIO-23-0993B) and the constructive comments of the reviewers. We enclose the revised manuscript, which addresses reviewer' remarks point-by-point. All changes in our manuscript are highlighted in yellow.

Sincerely,

Dan Zhu,

Ph.D., SPIE Fellow

Professor, Britton Chance Center for Biomedical Photonics, Huazhong University of Science and Technology

Deputy-Director, Wuhan National Laboratory for Optoelectronics

Secretary General & Vice-President, Biomedical Photonics Committee of Chinese Optical Society

Wuhan 430074, P.R. China

E-mail: dawnzh@mail.hust.edu.cn

Reviewer #3

Comment: In the revised manuscript by Liu et al., the authors have seriously taken my concerns into consideration, and performed additional experiments and analysis to support their main claims. Most of my questions are logically and evidently answered by the authors with new data, as well as citing related literatures in the field to support their conclusion. Also, updates that recognize the diversity of microglia were done. There only leave one point that I feel the author still did not address, which is listed below.

In my previous comment 1, I suggested the author to perform more causal studies rather than only reporting the correlation between phenotypes. While some part of the study it may be difficult to narrow down to a single target, and it is acceptable that the author provided one possible mechanism, like the IFN-gamma related part and the nNOS part, the major conclusion that touch microglia dysfunction should be causally revisited.

Based on the data in the first several figures, the author concluded in the title and abstract on their key finding, that reversing microglia dysfunction is the major reason (or at least one of the reason) for tPBM' s effect on diabetes. Logically, this can be written as tPBM · microglia · diabetes. I understand that microglia may not be the only reason, but current data did not support their importance, rather only provided correlation. The logic presented in the figure can be summarized (after adding diabetes phenotype) as tPBM · microglia, and tPBM · diabetes, which are parallel. This is what I comment as correlation in my previous question. In another word, tPBM induced microglia changes may not even have any contribution to the tPBM induced diabetes improvement, although I agree it is highly possible given microglia are so important and went wrong

during diseases. One possible experiment is to use loss of function on microglia, to see whether tPBM induced improvement on diabetes is affected. This could be achieved by using drugs to eliminate microglia, or to block P2Y12 receptors as the author suggested about their importance. Without such kind of experiments the key conclusion from the author is not well supported.

Response: We would like to express our sincere gratitude to the review for the continuous help in improving our paper. We have revised our abstract and the conclusions in the revised manuscript according to reviewer's suggestion.

Page 1, Lines 23-30: We find tPBM treatment effectively improves insulin therapy on microglial morphology and reactivity. We also show that tPBM stimulates brain drainage system through activation of meningeal lymphatics, which contributes to the removal of inflammatory factor, and increase of microglial purinergic receptor P2RY12. Besides, the energy expenditure and locomotor activity of diabetic mice are also improved by tPBM. Our results demonstrate that tPBM can be an efficient, non-invasive method for the treatment of microglial dysfunction caused by diabetes, and also has the potential to prevent diabetic physiological disorders.

Page 16, Lines 499-511: In conclusion, we clearly demonstrate that DM induces changes in microglial morphology and reactivity to cerebrovascular injury that is associated with the BBB leakage and inflammation of the brain tissues. The treatment of tPBM can effectively enhance the therapy of insulin on microglial morphology and function in diabetic mice. The mechanisms of therapeutic effects of tPBM are stimulation of brain drainage system, which improves the brain immunity through a decrease in the level of inflammatory factor IFN- γ , and an increase in the expression of microglial purinergic receptor P2RY12 in brain tissues that we observed in DM mice treated with tPBM+insulin. Given the important role of microglia in brain homeostasis and development, restoring microglial function may have the potential to ameliorate diabetic brain dysfunction as microglia are functionally abnormal during diabetes mellitus. But this needs to be further investigated in future work. Besides, we also found tPBM treatment could effectively improve the energy expenditure and locomotor activity in diabetic mice. These results demonstrate the potential of tPBM to treat diabetic microglial dysfunction and prevent physiological disorders in diabetes.